# Merge before Forget: A Single LoRA Continual Learning via Continual Merging

**Fuli Qiao**
Pennsylvania State University
`fvq5015@psu.edu`

**Mehrdad Mahdavi**
Pennsylvania State University
`mzm616@psu.edu`

## Abstract

Parameter-efficient continual learning has emerged as a promising approach for large language models (LLMs) to mitigate catastrophic forgetting while enabling adaptation to new tasks. Current Low-Rank Adaptation (LoRA) continual learning techniques often retain and freeze previously learned LoRAs or generate data representations to overcome forgetting, typically utilizing these to support new LoRAs learn new tasks. However, these methods not only ignore growing computational memory with tasks and limited storage space but also suffer from potential task interference due to the lack of effective LoRA merging mechanisms. In this paper, we propose a novel continual learning method that orthogonally initializes and sequentially merges LoRAs updates into a single unified LoRA. Our method leverages orthogonal basis extraction from previously learned LoRA to initialize the learning of new tasks, further exploits the intrinsic asymmetry property of LoRA components by using a time-aware scaling mechanism to balance new and old knowledge during continual merging. Our approach maintains constant memory complexity with respect to the number of tasks, minimizes interference between past and new tasks via orthogonal basis initialization, and improves performance over asymmetric LoRA merging via adaptive scaling. We provide theoretical analysis to justify our design and conduct extensive experiments across diverse continual learning benchmarks using various LLMs, demonstrating the effectiveness and efficiency of our method.

## 1 Introduction

Large Language Models (LLMs) (Raffel et al., 2020; Achiam et al., 2023; Touvron et al., 2023) have been growing as the cornerstone of modern machine learning, achieving remarkable performance across a wide range of downstream tasks. However, despite their impressive capabilities, LLMs still suffer from catastrophic forgetting (McCloskey & Cohen, 1989; Zhu et al., 2024a; Yang et al., 2024b) when fine-tuning sequential tasks, and their huge model capacity makes full fine-tuning computationally expensive and memory-intensive (Zhao et al., 2024a). These challenges have led to increasing attention in parameter-efficient continual learning, particularly via techniques such as LoRA (Low-Rank Adaptation) (Hu et al., 2022), that injects trainable low-rank matrices $\boldsymbol{A}$ and $\boldsymbol{B}$ into pre-trained models, enabling task adaptation with minimal additional parameters. Existing methods have shown progress in mitigating forgetting in LLMs through LoRA-based continual learning. For example, O-LoRA (Wang et al., 2023) freezes previously learned LoRAs and incrementally learns new tasks in their orthogonal subspace; InfLoRA (Liang & Li, 2024) preserves prior LoRAs and uses task-dependent input matrices to define orthogonal subspaces for initializing new ones; SAPT-LoRA (Zhao et al., 2024b) retains earlier LoRAs and leverages generated previous tasks' data to align new LoRA learning with shared modules; SD-LoRA (Wu et al., 2025) incrementally decouples the learning of magnitude and direction in LoRA components while preserving directions learned from previous tasks. However, these methods either keep and freeze previously learned LoRAs, resulting in parameter growth of the form $[\boldsymbol{B}_1\boldsymbol{A}_1, \ldots, \boldsymbol{B}_t\boldsymbol{A}_t]$, or generate and maintain task-specific data representations, leading to *(i) linear growth* in memory usage with the number of tasks, *(ii) limited scalability* due to constrained storage space, and *(iii) potential task interference* in the absence of principled LoRA merging mechanisms. These limitations motivate the question:

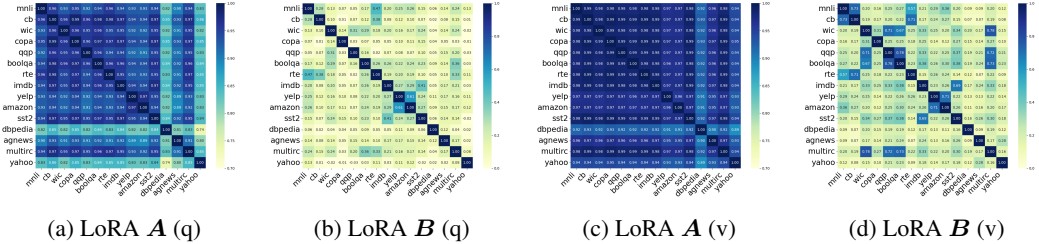

|(a) LoRA $A$ (q)|(b) LoRA $B$ (q)|(c) LoRA $A$ (v)|(d) LoRA $B$ (v)|

Figure 1: Cosine similarity between 15 tasks from the large number of tasks benchmark for fine-tuned q and v attention LoRA $A$ and $B$ in the last layer (32nd) of Llama-2-7B-chat.

*Can we enable continual learning only using a single shared LoRA, without learning or storing task-specific LoRAs or data representations?*

To address this question, we take inspiration from model merging (Garipov et al., 2018; Draxler et al., 2018; Wortsman et al., 2022), an emerging paradigm that aims to combine multiple task-specific models into a single unified model without retraining (Stoica et al., 2024; Ilharco et al., 2023; Yadav et al., 2023; Ortiz-Jimenez et al., 2023). By extending this idea, we frame continual learning as a sequential model merging problem, where the objective shifts from keeping all task-specific LoRAs or data to continually integrating their updates into a single shared LoRA as new tasks arrive. Recently, model merging has successfully been extended to the LoRA regime: KnOTS (Stoica et al., 2025) leverages singular value decomposition to project LoRA updates into a shared latent space, where existing merging methods can be applied; LoRA-LEGO (Zhao et al., 2025) decomposes LoRAs into minimal semantic units via grouping and clustering, enabling a reconstruction of multiple LoRAs into one. However, these LoRA merging methods generally assume *concurrent access* to all task-specific LoRAs fine-tuned from the same pre-trained model, which limits their applicability to the continual merging scenarios (Dziadzio et al., 2025), where tasks arrive sequentially. In such settings, the order of merging becomes critical and may degrade the performance of the final model. Moreover, continual LoRA merging remains underexplored in existing literature. While in the full-model setting, continual merging has received more attention, e.g., OPCM (Tang et al., 2025) sequentially projects new model updates onto subspaces orthogonal to the previously merged model and uses adaptive scaling to mitigate interference. However, these methods are not designed for LoRA, and the objective of merging differs from that of continual learning (Ortiz-Jimenez et al., 2023). These challenges lead to the following question we aim to answer:

*How can we enable continual learning through LoRA-based continual merging?*

We answer this question by maintaining a single pair of low-rank matrices $\{A, B\}$, shared across tasks. Achieving this necessitates addressing key challenges, including how to initialize and continually update the shared LoRA to effectively balance the trade-off between forgetting and generalization. Moreover, in contrast to full-model continual merging, $A$ and $B$ play different roles in continual merging with LoRA. For instance, prior works (Zhu et al., 2024b; Sun et al., 2024; Zhang et al., 2023b; Kopiczko et al., 2024) have shown that in LoRA fine-tuning, training $B$ (initialized to zero) is critical for the performance, even randomly initialized $A$ often suffices, but reversing the roles of $A$ and $B$ substantially decreases performance. To further investigate the asymmetry of LoRA components, we separately fine-tune 15 tasks from a standard large number of tasks benchmark (Wang et al., 2023) in continual learning using 15 independent LoRAs on Llama-2-7B-chat (Touvron et al., 2023), and compute cosine similarity of $A$ and $B$ across 15 tasks using their last layer LoRA. Figure 1 shows that $A$ exhibits significantly higher similarity across tasks compared to $B$, suggesting that LoRA components follow inherently different learning dynamics. This motivates us to treat $A$ and $B$ differently in continual merging.

To address the above questions, we propose a novel parameter-efficient continual learning method via continual merging into a single LoRA, which initializes new task learning in an orthogonal subspace and sequentially merges LoRA updates. We name this method **SLAO** (**S**ingle **L**oRA continual learning with **O**rthogonal initialization via continual merging). Specifically, SLAO initializes each new task learning LoRA using orthogonal basis extracted from previously learned LoRA components, and exploits the asymmetric roles of $A$ and $B$ by applying a time-aware scaling mechanism that balances knowledge retention and plasticity during continual merging. As shown in Figure 2, our approach ensures constant memory overhead regardless of the number of tasks. Additionally, it

reduces interference between past and new tasks via orthogonal basis initialization and enhances performance through adaptive continual merging that considers LoRA asymmetry.

**Summary of contributions.** This paper makes the following key contributions: (1) A novel parameter-efficient continual learning method for LLMs that continually merges new task LoRAs into a single LoRA via orthogonal basis initialization and a time-aware scaling mechanism, reducing catastrophic forgetting and improving generalization. (2) A theoretical analysis of how our design mitigates forgetting and improves intransigence. (3) Comprehensive experiments on various continual learning benchmarks using Llama models and Qwen models of varying sizes, demonstrating effectiveness and efficiency of our proposed method.

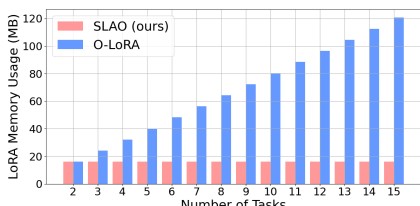

Figure 2: Comparison of SLAO and O-LoRA memory usage of large number of tasks benchmark via Llama-2-7B-chat.

## 2 RELATED WORKS

**Continual Learning.** Continual learning aims to retain knowledge of previously learned tasks while adapting to new data. It faces two main challenges: (1) catastrophic forgetting (McCloskey & Cohen, 1989), where the performance of the model on earlier tasks significantly degrades as it learns new ones; and (2) knowledge transfer, where the model leverages previously acquired knowledge to improve learning on new tasks. Existing approaches are divided into three categories:

(i) *Rehearsal-based methods* employ a memory buffer to store samples from previous tasks, enabling joint training with new tasks. Dark Experience Replay (Buzzega et al., 2020) encourages consistency with past knowledge by aligning the model's current logits with those sampled earlier in the optimization trajectory. CLEAR (Rolnick et al., 2019), an experience replay method, effectively mitigates catastrophic forgetting in multi-task reinforcement learning. Gradient episodic memory (Lopez-Paz & Ranzato, 2017) stores task-specific gradients and projects new gradients to avoid interference with previous knowledge.

(ii) *Regularization-based methods* utilize constraints on the parameters of the model to prevent model updates of new tasks from interfering with knowledge acquired on previous tasks. Elastic weight consolidation, EWC (Kirkpatrick et al., 2017), uses Fisher Information Matrix to identify and protect parameters critical for previous tasks. Orthogonal Gradient Descent, OGD (Farajtabar et al., 2020), projects gradients of new tasks onto a subspace that preserves model outputs on previous tasks, while ensuring the direction remains effective for learning new tasks.

(iii) *Architecture-based methods* dynamically adjust the structure of the model to isolate task-specific weights or expand model capacity (Mallya & Lazebnik, 2018; Wang et al., 2022b). Packnet (Mallya & Lazebnik, 2018) performs iterative pruning and network re-training. Progressive Prompts (Razdaibiedina et al., 2023) mitigate forgetting by maintaining a growing sequence of soft prompts, where each new task contributes an additional prompt.

**Parameter-efficient continual learning.** LoRA-based continual learning has emerged as a practical and parameter-efficient technique for adapting LLMs to sequential tasks. O-LoRA (Wang et al., 2023) freezes previously learned LoRAs and incrementally learns new tasks in their orthogonal subspace; InfLoRA (Liang & Li, 2024) preserves prior LoRAs and uses task-dependent input matrices to define orthogonal subspaces for initializing new ones; SAPT-LoRA (Zhao et al., 2024b) retains earlier LoRAs and leverages generated previous tasks' data to align new LoRA learning with shared modules; SD-LoRA (Wu et al., 2025) incrementally decouples the learning of magnitude and direction in LoRA components while preserving directions learned from previous tasks.

**Merging and Continual Merging.** Model merging (Garipov et al., 2018; Draxler et al., 2018; Wortsman et al., 2022) has emerged as an efficient paradigm that combines multiple task-specific models into a single unified model without retraining (Stoica et al., 2024; Ilharco et al., 2023; Yadav et al., 2023; Ortiz-Jimenez et al., 2023). This idea has recently extended to LoRA-based adaptation: KnOTS (Stoica et al., 2025) leverages singular value decomposition to project LoRA updates into a shared latent space, where existing merging methods can be applied; LoRA-LEGO (Zhao et al., 2025) decomposes LoRAs into minimal semantic units via grouping and clustering, enabling a reconstruc-

tion of multiple LoRAs into one. However, both LoRA merging and full model merging generally assume *simultaneous access to all task-specific LoRA or model* fine-tuned from the same initial pre-trained model, which limits their applicability to the continual merging scenarios (Dziadzio et al., 2025), where tasks arrive sequentially. Moreover, continual LoRA merging remains underexplored in existing literature. While in full-model settings, continual merging has received more attention, i.e., OPCM (Tang et al., 2025) mitigates interference by sequentially projecting new model updates onto subspaces orthogonal to the previously merged model, combined with adaptive scaling.

## 3 BACKGROUND AND MOTIVATION

**Problem setup.** Let $f_{\boldsymbol{W}_0} : \mathcal{X} \to \mathcal{Y}$ denote a pre-trained model parameterized by $\boldsymbol{W}_0 \in \mathbb{R}^{m \times n}$, which remains frozen throughout the continual learning (CL) process. Here, $\mathcal{X}$ and $\mathcal{Y}$ represent the input and output spaces, respectively. We consider a sequence of $T$ tasks. For each task $t \in \{1, 2, \ldots, T\}$, the model is continually fine-tuned using the LoRA algorithm based on its associated training dataset $\mathcal{D}_t = \{(\boldsymbol{X}_{t,i}, \boldsymbol{Y}_{t,i})\}_{i=1}^{N}$, and evaluated on a separate test dataset $\mathcal{D}'_t = \{(\boldsymbol{X}'_{t,i}, \boldsymbol{Y}'_{t,i})\}_{i=1}^{N'}$, where $N$ and $N'$ denote the number of training and testing samples, respectively. The goal is to continually learn a single set of LoRA parameters, specifically, matrices $\boldsymbol{B} \in \mathbb{R}^{m \times r}$ and $\boldsymbol{A} \in \mathbb{R}^{r \times n}$ with $r \ll \min(m, n)$ such that the resulting merged LoRA model remains competitive with models optimized for expected risk (multi-task objective):

$$\min_{\boldsymbol{B}, \boldsymbol{A}} \sum_{t=1}^{T} \sum_{(\boldsymbol{X}_t, \boldsymbol{Y}_t) \in \mathcal{D}'_t} \mathcal{L}_t(f_{\boldsymbol{W}_0 + \boldsymbol{B}\boldsymbol{A}}(\boldsymbol{X}_t), \boldsymbol{Y}_t), \tag{1}$$

where $\mathcal{L}_t$ denotes the empirical risk (e.g., cross-entropy or mean squared error) for task $t$.

In the single shared LoRA setting for CL, we are restricted to maintaining only one pair of LoRA parameters, denoted by $\boldsymbol{A}_{\text{merge}}^t$ and $\boldsymbol{B}_{\text{merge}}^t$, across all tasks. When task $t$ arrives, we fine-tune on its training data, possibly initialized with current merged models $\{\boldsymbol{A}_{\text{merge}}^{t-1}, \boldsymbol{B}_{\text{merge}}^{t-1}\}$, to obtain task-specific LoRA parameters $\boldsymbol{B}_{\text{ft},t} \in \mathbb{R}^{m \times r}$ and $\boldsymbol{A}_{\text{ft},t} \in \mathbb{R}^{r \times n}$. The fine-tuned model for task $t$ is represented as $f_{\boldsymbol{W}_0 + \boldsymbol{B}_{\text{ft},t} \boldsymbol{A}_{\text{ft},t}}(\cdot)$. After fine-tuning, we merge the previously accumulated LoRA parameters $\boldsymbol{B}_{\text{merge}}^{t-1} \boldsymbol{A}_{\text{merge}}^{t-1}$ with the new task-specific parameters $\boldsymbol{B}_{\text{ft},t} \boldsymbol{A}_{\text{ft},t}$, resulting in an updated merged representation $\boldsymbol{B}_{\text{merge}}^t \boldsymbol{A}_{\text{merge}}^t$. Due to the inherent asymmetry in LoRA components, the merging of the LoRA components is performed separately for $\boldsymbol{B}$ and $\boldsymbol{A}$ as formalized below:

$$\boldsymbol{B}_{\text{merge}}^t = \text{ContinualMerge\_B}(\boldsymbol{B}_{\text{merge}}^{t-1}; \boldsymbol{B}_{\text{ft},t}), \; \boldsymbol{A}_{\text{merge}}^t = \text{ContinualMerge\_A}(\boldsymbol{A}_{\text{merge}}^{t-1}; \boldsymbol{A}_{\text{ft},t}), \; t \geq 2$$

where $\boldsymbol{B}_{\text{ft},0} = \boldsymbol{0}$ and $\boldsymbol{A}_{\text{ft},0}$ is initialized using a Gaussian distribution, following the standard LoRA initialization (Hu et al., 2022). $\boldsymbol{B}_{\text{merge}}^1 = \boldsymbol{B}_{\text{ft},1}$ and $\boldsymbol{A}_{\text{merge}}^1 = \boldsymbol{A}_{\text{ft},1}$ are initialized as the first task fine-tuned LoRA, and $\boldsymbol{B}_{\text{merge}}^t \boldsymbol{A}_{\text{merge}}^t$ is to optimize Equation 1.

### 3.1 OPPORTUNITIES AND CHALLENGES IN CONTINUAL LEARNING VIA CONTINUAL MERGING

**Storage and memory efficiency** is one of the core advantages of continual merging for CL. Unlike existing continual learning methods that benefit from freezing and retaining previously fine-tuned LoRAs, using continual merging after fine-tuning task $t$ only requires storing a fixed number of LoRAs: (1) the current merged LoRA and (2) the fine-tuned LoRA to be merged. This strategy results in a constant memory complexity of $\mathcal{O}(|\boldsymbol{B}| + |\boldsymbol{A}|) = \mathcal{O}((m + n)r)$, where $|\boldsymbol{B}| + |\boldsymbol{A}|$ denote the parameter sizes of a single LoRA. Critically, this memory requirement remains independent of the number of sequential tasks $T$. In contrast, as shown in Figure 2, existing freezing-based continual learning methods require storing all LoRAs, incurring a linear memory complexity of $\mathcal{O}(T(|\boldsymbol{B}| + |\boldsymbol{A}|))$, which is $\mathcal{O}(T(m + n)r)$, growing linearly with the number of tasks.

**Training efficiency** is an evident advantage of continual merging for CL. Prior works do not simply keep previous tasks' LoRAs without any operations. Instead, when training new tasks, prior works use these multiple LoRAs during training through constraints, i.e., making new task LoRA parameters orthogonal to all previous LoRAs, heavily increasing computational cost during training. However, continual merging in CL would only use the parameters of a single previously fine-tuned LoRA to initialize new task LoRA parameters before training, avoiding extra computation during training.

**Difference between continual learning and merging** is mainly in the objective. In the context of

multi-task model merging, the task arithmetic property, as defined by Ortiz-Jimenez et al. (2023), refers to the ability to add task-specific vectors without interfering with performance on other tasks. However, in CL, the objective extends beyond retention: the model must both preserve previously acquired knowledge and generalize effectively to unseen data. Hence, while merging can support CL, its underlying objectives are not entirely equivalent to those of CL, leading to fundamental differences in both theoretical analysis and algorithmic design.

## 3.2 Orthogonal Initialization Motivated by LoRA NTK Analysis

To inform our algorithmic design, we evaluate the performance of CL using LoRA by two key metrics, forgetting error (Lin et al., 2023) and intransigence error (Li et al., 2023), defined as below:

(1) **Forgetting error:** It measures how much knowledge of old tasks has been forgotten after learning the current task. Specifically, after learning task $t \in [2, T]$, the average forgetting over all old tasks $i \in [1, t-1]$ is defined as:

$$\mathcal{F}_t = \sum_{i=1}^{t-1} \left( \mathcal{L}_i(\boldsymbol{W}_0 + \boldsymbol{B}_t \boldsymbol{A}_t) - \mathcal{L}_i(\boldsymbol{W}_0 + \boldsymbol{B}_i \boldsymbol{A}_i) \right) \tag{2}$$

In Equation 2, $\mathcal{L}_i(\boldsymbol{W}_0 + \boldsymbol{B}_t \boldsymbol{A}_t) - \mathcal{L}_i(\boldsymbol{W}_0 + \boldsymbol{B}_i \boldsymbol{A}_i)$ denotes the performance difference between $\boldsymbol{B}_i \boldsymbol{A}_i$ (result after training task $i$) and $\boldsymbol{B}_t \boldsymbol{A}_t$ (result after training task $t$) on test data of task $i$.

(2) **Intransigence error:** It evaluates the ability of the algorithm to adapt to a new task after having already adapted to a sequence of old tasks.

$$\mathcal{I}_t = \sum_{i=1}^{t} \left( \mathcal{L}_i(\boldsymbol{W}_0 + \boldsymbol{B}_i \boldsymbol{A}_i) - \mathcal{L}_i(\boldsymbol{W}_0 + \boldsymbol{B}_i^* \boldsymbol{A}_i^*) \right) \tag{3}$$

In Equation 3, $\mathcal{L}_i(\boldsymbol{W}_0 + \boldsymbol{B}_i \boldsymbol{A}_i) - \mathcal{L}_i(\boldsymbol{W}_0 + \boldsymbol{B}_i^* \boldsymbol{A}_i^*)$ denotes the performance difference between $\boldsymbol{B}_i^* \boldsymbol{A}_i^*$ (optimal result of training task $i$) and $\boldsymbol{B}_i \boldsymbol{A}_i$ (result after training task $i$) on test data of task $i$.

To examine these errors, we draw on the empirical observation (Malladi et al., 2023) that, when prompt-based fine-tuning is employed (Schick & Schütze, 2021; Gao et al., 2021), the fine-tuning of a pre-trained language model tends to remain within the Neural Tangent Kernel (NTK) regime. Specifically, under the NTK regime, assuming $\mathcal{D}_t = \{(\boldsymbol{X}_i, \boldsymbol{Y}_i)\}_{i \in \{1, \ldots, N\}}$, the empirical risk for task $t$ using LoRA can be approximated as (Jang et al., 2024)

$$\mathcal{L}_t = \frac{1}{N} \sum_{i=1}^{N} \ell_i \left( f_{\boldsymbol{W}_0}(\boldsymbol{X}_i) + \langle \nabla_{\boldsymbol{W}} f_{\boldsymbol{W}_0}(\boldsymbol{X}_i), \boldsymbol{B}_i \boldsymbol{A}_i \rangle, \boldsymbol{Y}_i \right) \tag{4}$$

As detailed in Appendix B (Lemma 1), by extending the analysis in Jang et al. (2024); Maurer (2016), we show that, under the NTK regime, the term in forgetting error can be bounded as follows:

$$\mathcal{L}_i(\boldsymbol{W}_0 + \boldsymbol{B}_t \boldsymbol{A}_t) - \mathcal{L}_i(\boldsymbol{W}_0 + \boldsymbol{B}_i \boldsymbol{A}_i) \leq G \| \langle \boldsymbol{B}_t \boldsymbol{A}_t - \boldsymbol{B}_i \boldsymbol{A}_i, \nabla_{\boldsymbol{W}} f_{\boldsymbol{W}_0}(\boldsymbol{X}_i) \rangle \|_2$$

$$\leq G \sqrt{\sum_{j=1}^{K} \|\boldsymbol{B}_t \boldsymbol{A}_t - \boldsymbol{B}_i \boldsymbol{A}_i\|_{\mathrm{F}}^2 \|\nabla_{\boldsymbol{W}} f_{\boldsymbol{W}_0}^{(j)}(\boldsymbol{X}_i)\|_{\mathrm{F}}^2} \leq G \sqrt{\sum_{j=1}^{K} \|\boldsymbol{B}_t \boldsymbol{A}_t - \boldsymbol{B}_i \boldsymbol{A}_i\|_{\mathrm{F}}^2 R^2} \tag{5}$$

where $K$ is output dimension and $\|\nabla_{\boldsymbol{W}} f_{\boldsymbol{W}_0}^{(j)}(\boldsymbol{X}_i)\|_{\mathrm{F}} \leq R$. Thus, to minimize Equation 2, we should make $\|\boldsymbol{B}_t \boldsymbol{A}_t - \boldsymbol{B}_i \boldsymbol{A}_i\|_{\mathrm{F}}$ as small as possible. Similarly, to minimize Equation 3, $\|\boldsymbol{B}_i \boldsymbol{A}_i - \boldsymbol{B}_i^* \boldsymbol{A}_i^*\|_{\mathrm{F}}$ should be minimized. Thus, the term in forgetting-intransigence decomposition can be written as:

$$\|\boldsymbol{B}_t \boldsymbol{A}_t - \boldsymbol{B}_i \boldsymbol{A}_i\|_{\mathrm{F}} + \|\boldsymbol{B}_i \boldsymbol{A}_i - \boldsymbol{B}_i^* \boldsymbol{A}_i^*\|_{\mathrm{F}}$$

$$\leq \|\boldsymbol{B}_t(\boldsymbol{A}_t - \boldsymbol{A}_i)\|_{\mathrm{F}} + \|(\boldsymbol{B}_t - \boldsymbol{B}_i)\boldsymbol{A}_i\|_{\mathrm{F}} + \|\boldsymbol{B}_i(\boldsymbol{A}_i - \boldsymbol{A}_i^*)\|_{\mathrm{F}} + \|(\boldsymbol{B}_i - \boldsymbol{B}_i^*)\boldsymbol{A}_i^*\|_{\mathrm{F}} \tag{6}$$

**Algorithmic motivation.** From above bound, we observe that forgetting-intransigence error in CL with LoRA depends asymmetrically on the choice of frozen and trainable components. For example, freezing $\boldsymbol{A}$ and fine-tuning $\boldsymbol{B}$ is at least as effective, if not better, than the reverse (Zhu et al., 2024b). However, if we apply freezing $\boldsymbol{A}$ in CL, then $\|\boldsymbol{A}_t - \boldsymbol{A}_i\|_{\mathrm{F}} = 0$ but $\|\boldsymbol{A}_i - \boldsymbol{A}_i^*\|_{\mathrm{F}}$ may be unintentionally increased due to random $\boldsymbol{A}_i$. Instead, if we propose to fine-tune $\boldsymbol{A}_i$ and extract orthogonal basis $\boldsymbol{Q}_{i-1}$ from $\boldsymbol{A}_{i-1}$, where $\boldsymbol{Q}_{i-1} \boldsymbol{Q}_{i-1}^\top = \boldsymbol{I}_r$, to initialize $\boldsymbol{A}_i^{(0)}$ via $\boldsymbol{Q}_{i-1}$, then we have $\boldsymbol{A}_i^{(0)}(\boldsymbol{A}_i^{(0)})^\top = \boldsymbol{I}_r$ where $i \in [1, \ldots, t]$. This orthogonal structure not only keeps geometric consistency across tasks but also allows $\boldsymbol{A}_j$ $(t \geq j > i)$, to remain well-aligned with previous $\boldsymbol{A}_i$, i.e. $\mathbb{E}[\boldsymbol{A}_j \boldsymbol{A}_i^\top] \approx \boldsymbol{I}_r$, thereby minimizing both $\|\boldsymbol{A}_t - \boldsymbol{A}_i\|_{\mathrm{F}}$ and $\|\boldsymbol{A}_i - \boldsymbol{A}_i^*\|_{\mathrm{F}}$. This motivates our design of orthogonal initialization. The complete derivation is in Appendix B.1.

### 3.3 Continual merging motivated by LoRA asymmetry analysis

To provide the analysis of merging $\boldsymbol{B}$, we consider a scenario where a single LoRA is continually fine-tuned for sequential tasks, which means each task starts from the previous task's fine-tuned LoRA. We initialize task 1 as $\boldsymbol{B}_0 = 0$ and $\boldsymbol{A}_0 \sim \mathcal{N}(0, \sigma^2)$ (Hu et al., 2022). After fine-tuning with $T$ steps, we obtain its parameters:

$$\boldsymbol{W}_0 + (\boldsymbol{B}_0 + \Delta\boldsymbol{B}_1)(\boldsymbol{A}_0 + \Delta\boldsymbol{A}_1) = \boldsymbol{W}_0 + \Delta\boldsymbol{B}_1\boldsymbol{A}_0 + \Delta\boldsymbol{B}_1\Delta\boldsymbol{A}_1 \tag{7}$$

Since $\boldsymbol{B}_0 = 0$, we have that

$$\|(\Delta\boldsymbol{B}_1)^\top \boldsymbol{B}_0\|_F = 0, \quad \|\boldsymbol{A}_0(\Delta\boldsymbol{A}_1)^\top\|_F \neq 0 \tag{8}$$

Based on theorem in Hao et al. (2024), we write task 1 fine-tuned LoRA:

$$\boldsymbol{B}_1 = \eta f_B(T)\boldsymbol{A}_0^\top, \quad \boldsymbol{A}_1 = \boldsymbol{A}_0 + \eta\boldsymbol{A}_0 f_A(T) \tag{9}$$

Using recursion, for task $i$ where $i \geq 2$, the orthogonality measures become:

$$\|\Delta\boldsymbol{B}_i^\top \boldsymbol{B}_{i-1}\|_F \approx \|(\eta f_B(T)\boldsymbol{A}_{i-1}^\top)^\top(\eta f_B(T)\boldsymbol{A}_{i-2}^\top)\|_F = \eta^2\|\boldsymbol{A}_{i-1}f_B(T)^\top f_B(T)\boldsymbol{A}_{i-2}^\top\|_F \tag{10}$$

$$\|\boldsymbol{A}_{i-1}\Delta\boldsymbol{A}_i^\top\|_F \approx \|(\boldsymbol{A}_{i-2} + \eta\boldsymbol{A}_{i-2}f_A(T))(\eta\boldsymbol{A}_{i-1}f_A(T))^\top\|_F$$
$$= \eta\|\boldsymbol{A}_{i-2}f_A(T)^\top\boldsymbol{A}_{i-1}^\top + \eta \cdot \boldsymbol{A}_{i-2}f_A(T)f_A(T)^\top\boldsymbol{A}_{i-1}^\top\|_F \tag{11}$$

Our insight is that the second term in Equation 11 has a smaller magnitude when learning rate is not large, since when $\eta \ll 1/L$, $\lim_{t\to\infty} \eta\|f_A(t)\| \ll 1$ (Hao et al., 2024), thus the second term is significantly smaller than the first term, and $\|\Delta\boldsymbol{B}_i^\top \boldsymbol{B}_{i-1}\|_F < \|\boldsymbol{A}_{i-1}\Delta\boldsymbol{A}_i^\top\|_F$. Hence, the update of $\boldsymbol{B}$ is more orthogonal to its initialization than the update of $\boldsymbol{A}$ is to its initialization. Based on the findings in Wei et al. (2025), task vectors in model merging are inherently close orthogonal to minimize interference, which indicates that in our case, merging $\boldsymbol{B}$ rather than merging $\boldsymbol{A}$ provides better task isolation and reduced interference, motivating our choice to perform merging $\boldsymbol{B}$.

Then, for the operation of merging $\boldsymbol{B}$, we build on parameter-efficient module linear arithmetic composition, including addition and negation. Linear connectivity implies that model parameters fine-tuned from the same pretrained checkpoint can be added to improve generalization (Wortsman et al., 2022), a property that extends to PEFT adapters, whose small updates likewise allow linear composition (Zhang et al., 2023a). Hence, we can write merging operations on $\boldsymbol{B}$ using task vectors

$$\boldsymbol{B}_{merge} = \boldsymbol{B}_{merge} + \lambda \cdot (\boldsymbol{B}_{new} - \boldsymbol{B}_{merge}) \tag{12}$$

This makes the foundation of the operation of merging $\boldsymbol{B}$ in continual merging.

## 4 Methodology

### 4.1 SLAO: Single LoRA Continual Learning

Based on the above analysis, we propose our method, SLAO, to utilize continual merging into a single LoRA to minimize task interference and improve generalization for CL. SLAO is motivated by four key insights: (1) *Orthogonal Retention:* To minimize forgetting error and intransigence error, it is crucial to maintain orthogonality in the LoRA components across tasks; (2) *Continual Merging:* To reduce memory usage in LoRA-based CL, continually merging new task fine-tuned LoRA updates into a single merged LoRA is a highly efficient strategy; (3) *Asymmetry of LoRA:* Given the distinct learning roles of LoRA components $\boldsymbol{A}$ and $\boldsymbol{B}$, they should be handled separately; and (4) *Time-aware Scaling:* To retain prior knowledge while adapting to new tasks, the merging process for new LoRA updates should be scaled in a time-aware manner that reflects its training trajectory.

The SLAO consists of two main operations: (1) Initialize each new task learning LoRA by extracted orthogonal basis from the previous task's fine-tuned LoRA; (2) After fine-tuning on the new task, we utilize the asymmetry of LoRA components to employ adaptive time-varying scaling for new LoRA updates to merge into the merged LoRA. The complete procedure is outlined in Algorithm 1 and illustrated in Figure 4. Starting with the first task's fine-tuned LoRA $\boldsymbol{B}_{ft,1}\boldsymbol{A}_{ft,1}$ by standard fine-tuning, our method iteratively integrates LoRA updates of subsequent tasks in a continual fashion.

**Orthogonal basis extraction for initialization.** For new task $i$, we first extract orthogonal basis of

---

**Algorithm 1** SLAO: Single LoRA Continual Learning

---

1: Initialize $\boldsymbol{B}_{\text{merge}}^1 = \boldsymbol{B}_{\text{ft},1}$, $\boldsymbol{A}_{\text{merge}}^1 = \boldsymbol{A}_{\text{ft},1}$, scaling factor $\lambda(1) = 1$, number of tasks $T$.
2: **for** $i = 2$ to $T$ **do**
3: $\quad\boldsymbol{Q}_i\boldsymbol{R}_i = QR((\boldsymbol{A}_{\text{ft},i-1})^\top), \boldsymbol{Q}_i = \boldsymbol{Q}_i \cdot \text{sign}(\text{diag}(\boldsymbol{R}_i))^\top$ // Extract orthogonal basis of $\boldsymbol{A}_{\text{ft},i-1}$
4: $\quad\boldsymbol{A}_{\text{ft},i}^{(0)} = \boldsymbol{Q}_i^\top, \boldsymbol{B}_{\text{ft},i}^{(0)} = \boldsymbol{B}_{\text{ft},i-1}$ // **Initialize** $\boldsymbol{A}_{\text{ft},i}$ and $\boldsymbol{B}_{\text{ft},i}$ for task $i$
5: $\quad\boldsymbol{B}_{\text{ft},i}\boldsymbol{A}_{\text{ft},i} \leftarrow \text{fine-tune}(\boldsymbol{W}_0, \boldsymbol{B}_{\text{ft},i}^{(0)}\boldsymbol{A}_{\text{ft},i}^{(0)})$ // **Fine-tune** $\boldsymbol{A}_{\text{ft},i}$ and $\boldsymbol{B}_{\text{ft},i}$ for task $i$
6: $\quad\boldsymbol{A}_{\text{merge}}^i = \boldsymbol{A}_{\text{ft},i}$
7: $\quad\boldsymbol{B}_{\text{merge}}^i = \boldsymbol{B}_{\text{merge}}^{i-1} + \lambda(i)(\boldsymbol{B}_{\text{ft},i} - \boldsymbol{B}_{\text{merge}}^{i-1})$
8: $\quad$Use merged LoRA $\boldsymbol{B}_{\text{merge}}^i\boldsymbol{A}_{\text{merge}}^i$ for inference until new task comes
9: **end for**
10: **return** $\boldsymbol{B}_{\text{merge}}^T\boldsymbol{A}_{\text{merge}}^T$

---

previous fine-tuned $\boldsymbol{A}_{\text{ft},i-1}$, use that to initialize $\boldsymbol{A}_{\text{ft},i}^{(0)}$, making $\boldsymbol{A}_{\text{ft},i}^{(0)}(\boldsymbol{A}_{\text{ft},i}^{(0)})^\top = \boldsymbol{I}_r$. We utilize QR decomposition to extract orthogonal matrix from $\boldsymbol{A}_{\text{ft},i-1}$, which is:

$$\boldsymbol{Q}_i\boldsymbol{R}_i = QR((\boldsymbol{A}_{\text{ft},i-1})^\top) \ \rightarrow \ \boldsymbol{Q}_i = \boldsymbol{Q}_i \cdot \text{sign}(\text{diag}(\boldsymbol{R}_i))^\top \ \rightarrow \ \boldsymbol{A}_{\text{ft},i}^{(0)} = \boldsymbol{Q}_i^\top \tag{13}$$

As a result, the initialization $\boldsymbol{A}_{\text{ft},i}^{(0)}$ has orthogonal rows. For $\boldsymbol{B}$, we directly initialize $\boldsymbol{B}_{\text{ft},i}^{(0)}$ by $\boldsymbol{B}_{\text{ft},i-1}$ which is the fine-tuned $\boldsymbol{B}$ of previous task $i-1$.

**Asymmetrically merging LoRA via time-aware scaling.** After fine-tuning task $i$, we merge its LoRA updates into $\{\boldsymbol{B}_{\text{merge}}^{i-1}, \boldsymbol{A}_{\text{merge}}^{i-1}\}$. Due to the intrinsic asymmetry of $\boldsymbol{B}$ and $\boldsymbol{A}$ in LoRA, we update $\boldsymbol{A}_{\text{merge}}^i = \boldsymbol{A}_{\text{ft},i}$, and we merge $\boldsymbol{B}$ by time-aware coefficient $\lambda(i)$ for new task updates:

$$\boldsymbol{B}_{\text{merge}}^i = \boldsymbol{B}_{\text{merge}}^{i-1} + \lambda(i) \cdot (\boldsymbol{B}_{\text{ft},i} - \boldsymbol{B}_{\text{merge}}^{i-1}) \tag{14}$$

where $\lambda(i)$ is introduced to maintain a consistent magnitude of the merged $\boldsymbol{B}$'s deviation from previous tasks throughout the merging process. In our method, the scaling factor can be set to $\lambda(i) = \frac{1}{\sqrt{i}}$, which follows the continual merging method proposed in Tang et al. (2025). The findings in Ilharco et al. (2023); Tang et al. (2024) indicate that task vectors from different tasks tend to be approximately orthogonal, and since $\boldsymbol{B}$s across tasks are approximately orthogonal to each other, as shown in Figure 1, which indicates that $\boldsymbol{B}$ task vectors are approximately orthogonal. This orthogonality makes $\lambda(i) = \frac{1}{\sqrt{i}}$ a natural choice for the scaling factor, since it helps maintain the magnitude of parameter changes across merging steps Tang et al. (2025).

## 4.2 Dynamics of SLAO

To better understand the effectiveness of SLAO, inspired by the analysis of Hao et al. (2024), in the following theorem we analyze the dynamics of task-specific parameters' update in CL scenario.

**Theorem 1.** *Let the parameters $\boldsymbol{A}$ and $\boldsymbol{B}$ be updated using SGD at each step $s$ for task $i$ as follows:*

$$\boldsymbol{A}_i^{s+1} = \boldsymbol{A}_i^s - \eta(\boldsymbol{B}_i^s)^\top(\nabla_{\boldsymbol{W}}\mathcal{L}_i^s), \ \ \boldsymbol{B}_i^{s+1} = \boldsymbol{B}_i^s - \eta(\nabla_{\boldsymbol{W}}\mathcal{L}_i^s)(\boldsymbol{A}_i^s)^\top \tag{15}$$

*where $\eta$ is the learning rate. We assume $\boldsymbol{A}_i^s = \boldsymbol{A}_i^{(0)} + \eta\boldsymbol{A}_i^{(0)}f_A(s)$ and $\boldsymbol{B}_i^s = \boldsymbol{B}_i^{(0)} + \eta f_B(s)(\boldsymbol{A}_i^{(0)})^\top$ holds with such functions $f_A$ and $f_B$ for $1, \ldots, s$, and $\|\sum_{s=1}^S \nabla_{\boldsymbol{W}}\mathcal{L}_i^{(s)}\|_{\text{F}} \le L$ for every $S$ during training task $i$, which implies that the model stays within a finite Euclidean ball. If we assume $\boldsymbol{A}_i^{(0)}(\boldsymbol{A}_i^{(0)})^\top = \boldsymbol{I}_r$, in this case, the dynamics of $\boldsymbol{A}_i$ satisfies $\|f_A(s)\|_2 \le \frac{\eta L^2(1-(\eta^2 L^2)^s)}{1-\eta^2 L^2}$, and the dynamics of $\boldsymbol{B}$ satisfies $f_B(s) = -\sum_{j=0}^{s-1}(\nabla_{\boldsymbol{W}}\mathcal{L}_i^j)(\eta f_A^\top(j) + \boldsymbol{I})$. When $\eta$ is small, we have $f_B(s) \approx -\sum_{j=0}^{s-1}(\nabla_{\boldsymbol{W}}\mathcal{L}_i^j)$. Thus $\boldsymbol{B}_i^S = \eta f_B(S)(\boldsymbol{A}_i^{(0)})^\top$, and total update for $\boldsymbol{B}_i$ is $\Delta\boldsymbol{B}_i = -\eta\left(\sum_{s=0}^S(\nabla_{\boldsymbol{W}}\mathcal{L}_i^s)\right)(\boldsymbol{A}_i^{(0)})^\top$.*

The proof is deferred to Appendix B. This analysis, under the orthogonal initialization of $\boldsymbol{A}$, suggests that $\boldsymbol{B}$ may update across different initialization subspaces, effectively increasing the rank of $\boldsymbol{B}$ and thereby aiding generalization. We note that the key difference between ours and Hao et al. (2024) lies in the initialization of LoRA: while they use standard initialization with $\boldsymbol{B}_i^{(0)} = \boldsymbol{0}$, we initialize $\boldsymbol{B}$ using the previously fine-tuned LoRA parameters, resulting in $\boldsymbol{B}_i^{(0)} \ne \boldsymbol{0}$, complicating the analysis.

## 5 EXPERIMENTS

### 5.1 EXPERIMENTAL SETUP

**Models and datasets.** We evaluate our approach across three Llama models: Llama-2-7B-chat, Llama-2-13B-chat, and Llama-3-2-3B, and two Qwen models: Qwen2.5-3B and Qwen2.5-7B. All experiments are conducted on NVIDIA A100 GPUs utilizing DeepSpeed repository. We consider three continual learning benchmarks: (1) *Standard CL benchmark*: AG News, Amazon, Reviews, Yelp Reviews, DBpedia, and Yahoo Answers. (2) *Large number of tasks*: five standard CL benchmark tasks, four GLUE tasks (MNLI, QQP, RTE, SST-2), five SuperGLUE tasks (WiC, CB, COPA, MultiRC, BoolQ), and IMDB movie reviews. Following O-LoRA (Wang et al., 2023), each task uses 1000 randomly sampled training samples and 500 validation samples per class. (3) *SuperNI Benchmark* (Wang et al., 2022a): A diverse collection of NLP tasks with expert-written instructions, covering dialogue generation, information extraction, question answering, summarization, and sentiment analysis. We follow task selection and ordering in SAPT (Zhao et al., 2024b), using 1,000 training instances and 100 for validation/testing per task.

**Baselines.** We compare our method SLAO with the following baselines: (1) *Continual learning baselines:* SeqLoRA: sequentially fine-tunes a single LoRA on multiple tasks without constraints; IncLoRA: incrementally adds a new LoRA per task while freezing previous LoRAs; O-LoRA (Wang et al., 2023); InfLoRA (Liang & Li, 2024); SAPT-LoRA (Zhao et al., 2024b);MTL: a single model is trained jointly on all tasks; LoRM (Salami et al., 2025); CorDA (knowledge-preserved adaptation) (Yang et al., 2024b); Magmax (Marczak et al., 2024). (2) *LoRA merging baselines:* LoRA-LEGO (Zhao et al., 2025); KnOTS (Stoica et al., 2025). (3) *Continual merging baseline:* OPCM (Tang et al., 2025). To fairly evaluate existing merging methods in LoRA-based continual learning, we extend full-model merging methods to LoRA and equally treat components of LoRA, and all merging methods are achieved sequentially.

**Evaluation metrics.** To evaluate our proposed approach, we employ three key metrics: (1) average accuracy (AA), calculated as the mean accuracy across all tasks after training on the last task: $\frac{1}{T}\sum_{i=1}^{T} a_{i,T}$, where $a_{i,T}$ is accuracy for classification tasks and Rouge-L for other tasks; (2) backward transfer (BWT) (Lin et al., 2022), defined as $\frac{1}{T-1}\sum_{i=1}^{T-1}(a_{i,T} - a_{i,i})$, and experimental results are shown in Appendix D.4; (3) maximum order-normalized performance disparity (MOPD) and average order-normalized performance disparity (AOPD) (Yoon et al., 2020), which evaluate order-robustness, and experimental results are shown in Appendix D.5.

### 5.2 OVERALL RESULTS

**Continual learning performance results analysis.** As shown in Table 1, our method consistently outperforms all data-free baselines across three benchmarks using Llama-2-7B-chat. *LoRA-Based continual learning:* SeqLoRA performs worst among LoRA-based methods, as unconstrained continual fine-tuning on a single LoRA causes severe forgetting. IncLoRA improves by freezing prior learned LoRAs to isolate subspaces, though its subspace separation is simple. InfLoRA outperforms O-LoRA in standard CL benchmark due to orthogonal input-based subspaces, but drops on large number of tasks and SuperNI benchmark, due to sensitivity to manually tuned DualGPM threshold. SAPT-LoRA achieves the highest average performance among LoRA-based methods, but relies on generated previous task pseudo samples, unrealistic in many LLM scenarios, and is more order-sensitive than ours. LoRM-BA (begin with freezing $B$) and LoRM-AB yield nearly identical results, suggesting that the freezing order of LoRA components in CL matters little. CorDA performs well on standard CL benchmark and large number of tasks, but drops significantly on SuperNI benchmark, likely due to relying on nullspace selection from pretrained models and lacking time-aware merging. MagMax performs comparably to ours on standard CL benchmark, slightly worse on large number of tasks, but underperforms on SuperNI benchmark, where task similarity is lower, thus only keeping weights which have the largest absolute value would cause forgetting. *LoRA merging baselines:* KnOTS and LoRA-LEGO perform similarly in the standard CL benchmark, but KnOTS outperforms in the large number of tasks and SuperNI. KnOTS may benefit from flexible SVD-merging mechanism so that we apply time-aware scaling on merging, while LoRA-LEGO treats tasks equally, lacks prioritization, and is ineffective in complex CL contexts. *Continual merging approaches:* since OPCM is designed for full model, directly applying it to LoRA by treating its

Table 1: Testing performance (%) on three CL benchmarks using Llama-2-7B-chat across different task orders, where each result is run three random times, where $Oi$ denotes $i$th task order.

| Method | Standard CL Benchmark | | | | Large Number of Tasks | | | | SuperNI Benchmark | | |
|---|---|---|---|---|---|---|---|---|---|---|---|
| | O1 | O2 | O3 | avg | O4 | O5 | O6 | avg | O1 | O2 | avg |
| SeqLoRA | 73.3 | 76.2 | 78.4 | 76.0 | 69.1 | 66.0 | 71.1 | 68.7 | 18.4 | 26.8 | 22.6 |
| IncLoRA | 75.3 | 77.3 | 78.3 | 77.0 | 72.2 | 71.6 | 73.8 | 72.5 | 22.0 | 25.6 | 23.8 |
| O-LoRA | 76.1 | 76.3 | 79.2 | 77.2 | 74.0 | 72.0 | 74.6 | 73.5 | 23.3 | 28.4 | 25.9 |
| InfLoRA | 78.4 | 80.4 | 79.9 | 79.6 | 69.4 | 67.4 | 72.5 | 69.8 | 16.5 | 22.1 | 19.3 |
| SPAT-LoRA | 82.9 | 81.8 | 78.7 | 81.1 | 84.7 | 78.9 | 82.2 | 81.9 | 53.2 | 48.5 | 50.9 |
| LoRM-BA | 76.0 | 76.8 | 78.3 | 77.0 | 71.4 | 69.0 | 70.3 | 70.2 | 25.6 | 18.7 | 22.2 |
| LoRM-AB | 77.5 | 74.7 | 75.9 | 76.0 | 71.0 | 69.5 | 70.2 | 70.2 | 25.6 | 23.7 | 24.7 |
| CorDA | 78.4 | 79.3 | 80.0 | 79.2 | 73.4 | 72.7 | 74.0 | 73.4 | 20.9 | 16.0 | 18.5 |
| MagMax | 80.1 | 80.6 | 80.3 | 80.3 | 72.3 | 73.5 | 74.5 | 73.4 | 15.3 | 7.0 | 11.2 |
| KnOTS | 67.9 | 65.9 | 70.8 | 68.2 | 61.5 | 60.1 | 58.0 | 59.9 | 34.6 | 30.1 | 32.4 |
| LoRA-LEGO | 68.3 | 66.0 | 70.9 | 68.4 | 58.8 | 58.7 | 53.2 | 56.9 | 32.8 | 26.7 | 29.8 |
| OPCM | 61.9 | 62.0 | 56.7 | 60.2 | 51.9 | 52.8 | 46.9 | 50.5 | 11.6 | 12.3 | 12.0 |
| SLAO (ours) | 80.1 | 80.8 | 80.4 | 80.4 | 75.0 | 74.4 | 75.1 | 74.8 | 38.7 | 35.7 | 37.2 |
| Multi-Task | | 80.9 | | | | 78.1 | | | | 45.2 | |

Table 2: Comparison of initialization strategies on testing performance across three standard CL benchmarks using Llama-2-7B-chat under different task orders, where $Oi$ denotes $i$th task order.

| Initialization | Standard CL Benchmark | | | | Large Number of Tasks | | | | SuperNI Benchmark | | |
|---|---|---|---|---|---|---|---|---|---|---|---|
| | O1 | O2 | O3 | avg | O4 | O5 | O6 | avg | O1 | O2 | avg |
| Random (zero) | 66.4 | 62.4 | 68.4 | 65.7 | 61.4 | 60.3 | 57.2 | 59.6 | 33.3 | 28.9 | 31.1 |
| Last-Merge | 80.1 | 80.8 | 80.1 | 80.3 | 74.7 | 72.8 | 75.0 | 74.2 | 37.4 | 30.5 | 34.0 |
| Last-FT (ours) | 80.1 | 80.8 | 80.4 | 80.4 | 75.0 | 74.4 | 75.1 | 74.8 | 38.7 | 35.7 | 37.2 |

two components identically leads to suboptimal performance. The results under Llama-2-13B-chat, Qwen2.5-3B, and Qwen2.5-7B (Yang et al., 2024a) are shown in Appendix D.

**Impact of initialization strategies.** We compare three different initialization strategies for learning new tasks: (1) random (zero) initialization, (2) initialization from last merging point, (3) initialization from last fine-tuning point (ours). As shown in Table 2, initializing from last fine-tuning point consistently outperforms other two strategies across all three benchmarks. Using last merging point performs slightly worse, while random (zero) initialization performs the worst. The performance gap is due to how initialization affects LoRA's learning trajectory and merging way. Random initialization places $A$ far away from optimal task-specific $A^*$, making intransigence worse. Initialization from last merging point fixes time coefficients after merging back to a single LoRA, limiting its flexibility, while initialization from last fine-tuning point allows the merged LoRA to implicitly reweight previous tasks' updates when merging. This adaptive adjustment yields better CL performance.

**Asymmetry in LoRA merging.** To investigate asymmetry in LoRA merging, we compare different continual merging strategies for LoRA: (1) Freeze A Merge B (FREA-MB), (2) Freeze B Merge A (FREB-MA), (3) Fine-tune BA Merge A (FTBA-MA), (4) Fine-tune BA Merge BA (FTBA-MBA), (5) Fine-tune BA Merge B (FTBA-MB). As shown in Table 3, only FTBA-MB consistently outperforms other strategies, except ours. FTBA-MBA has comparable performance compared to FTBA-MB, but FTBA-MA yields the poorest performance among fine-tuning LoRA methods. When freezing one component of LoRA, FREA-MB is better than FREB-MA, consistent with Zhu et al. (2024b) that freezing $A$ and fine-tuning $B$ is at least better than the reverse. It highlights asymmetry in LoRA components and importance of asymmetric merging based on their fundamental roles in adaptation.

**Effect of model variants and sizes.** We evaluate our method on three LLM variants: Llama-2-7B-chat, Llama-2-13B-chat, Llama-3-3B (Grattafiori et al., 2024). The results in Table 4 indicate that

Table 3: Comparison of merging strategies on testing performance on three standard CL benchmarks using Llama-2-7B-chat across different task orders, where $Oi$ denotes $i$th task order.

| Merging | Standard CL Benchmark | | | | Large Number of Tasks | | | | SuperNI Benchmark | | |
|---|---|---|---|---|---|---|---|---|---|---|---|
| | O1 | O2 | O3 | avg | O4 | O5 | O6 | avg | O1 | O2 | avg |
| FREB-MA | 77.7 | 78.0 | 76.2 | 77.3 | 70.8 | 66.1 | 72.1 | 69.7 | 13.9 | 25.4 | 19.7 |
| FREA-MB | 78.7 | 79.3 | 78.0 | 78.7 | 72.3 | 73.0 | 73.5 | 72.9 | 23.7 | 29.2 | 26.5 |
| FTBA-MA | 76.2 | 79.0 | 79.9 | 78.4 | 71.6 | 69.5 | 74.1 | 71.7 | 21.1 | 30.7 | 25.9 |
| FTBA-MBA | 79.7 | 80.4 | 80.2 | 80.1 | 73.2 | 73.9 | 74.5 | 73.9 | 33.8 | 32.7 | 33.3 |
| FTBA-MB | 79.3 | 80.8 | 80.1 | 80.1 | 74.1 | 74.0 | 74.8 | 74.3 | 32.4 | 35.2 | 33.8 |
| SLAO (ours) | 80.1 | 80.8 | 80.4 | 80.4 | 75.0 | 74.4 | 75.1 | 74.8 | 38.7 | 35.7 | 37.2 |

Table 4: Comparison of model variants and sizes on testing performance across three standard CL benchmarks using Llama-2-7B-chat in different task orders, where $Oi$ denotes $i$th task order.

| Model | Standard CL Benchmark | | | | Large Number of Tasks | | | | SuperNI Benchmark | | |
|---|---|---|---|---|---|---|---|---|---|---|---|
| | O1 | O2 | O3 | avg | O4 | O5 | O6 | avg | O1 | O2 | avg |
| Llama-3-2-3B | 74.3 | 75.8 | 75.3 | 75.1 | 73.3 | 72.5 | 74.9 | 73.6 | 32.7 | 34.6 | 33.7 |
| Llama-2-7B-chat | 80.1 | 80.8 | 80.4 | 80.4 | 75.0 | 74.4 | 75.1 | 74.8 | 38.7 | 35.7 | 37.2 |
| Llama-2-13B-chat | 80.8 | 81.1 | 81.1 | 81.0 | 76.5 | 75.9 | 76.0 | 76.1 | 42.3 | 42.2 | 42.3 |

model variants and model size play crucial roles in average performance across different task orders and benchmarks. Llama-3-3B performs the worst, while in same generation of Llama, larger models consistently achieve better accuracy: Llama-2-13B-chat relatively has higher accuracy compared to Llama-2-7B-chat. This trend suggests that increased model capacity enhances both reducing catastrophic forgetting and improving generalization in continual learning. Moreover, we observe that larger models exhibit greater robustness to task order variations compared to smaller models.

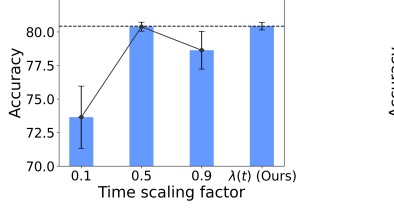
(a) Standard CL Benchmark-7B

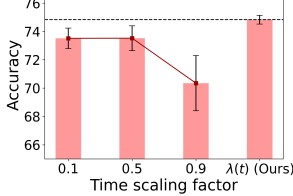
(b) Large Number of tasks-7B

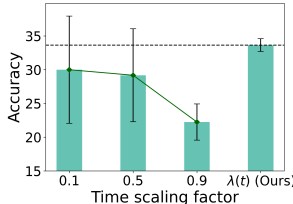
(c) SuperNI Benchmark-3B

Figure 3: Comparison of model performance across time coefficients.

**Time-varying coefficient analysis.** To evaluate impact of adaptive time-varying scaling in continual merging for CL, we compare it against fixed factors $\{0.1, 0.5, 0.9\}$ on three benchmarks via Llama-2-7B-chat and Llama-3-2-3B. As shown in Figure 3, adaptive strategy consistently achieves highest average accuracy with lower variance across task orders and models. For simpler standard CL benchmark, larger fixed value $0.9$ outperforms smaller one $0.1$, while for more complex or long benchmarks, smaller values perform better; $0.5$ is relatively stable but consistently suboptimal.

## 6 CONCLUSION

In this work, we proposed a novel parameter-efficient continual learning based on continual merging of LoRA, enabling no additional training or access to any data representations. Our approach leverages the orthogonal basis from previous fine-tuned LoRA to initialize for new task learning and constructs a single shared merged LoRA via time-aware scaling, thus ensuring constant memory usage regardless of task number. Through comprehensive experiments, we demonstrated the effectiveness and efficiency of our method across multiple benchmarks and model scales.

ACKNOWLEDGMENT

This work was partially supported by NSF EFMA Award #2318101 and NSF CAREER Award #2239374.

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

APPENDIX

The appendix is organized as follows:

## A  THE USE OF LARGE LANGUAGE MODELS (LLMs)

We utilize LLMs to polish the paper writing.

## B  THEORETICAL ANALYSIS

### B.1  THE FORGETTING-INTRANSIGENCE ERROR DECOMPOSITION UNDER NTK

**Lemma 1.** *(Jang et al., 2024; Maurer, 2016) Assume $\mathcal{D}$ is $i.i.d$ $N$ random samples sampled from probability distribution $\mathcal{P}$. Let $\mathcal{A}_D = \{\boldsymbol{X}_i \to f_{\boldsymbol{W}_0}(\boldsymbol{X}_i) + \langle \nabla_{\boldsymbol{W}} f_{\boldsymbol{W}_0}(\boldsymbol{X}_i), \boldsymbol{\delta} \rangle \in \mathbb{R}^K : \|\boldsymbol{\delta}\|_* \leq D, \boldsymbol{\delta} \in \mathbb{R}^{m \times n}\}$ is class of affine predictors with bounded nuclear norm $D$. For $1 \leq j \leq K$, suppose $\|\nabla_{\boldsymbol{W}} f_{\boldsymbol{W}_0}^{(j)}(\boldsymbol{X})\|_{\mathrm{F}} \leq R$ almost surely with respect to the random data $\boldsymbol{X}_i \sim \mathcal{P}$. For $1 \leq i \leq N$, suppose $\ell_i = \ell(\cdot, \boldsymbol{Y}_i)$ is $G$-Lipschitz continuous on $\mathcal{A}$ on the first argument (with respect to the Euclidean norm) for almost surely with respect to the random data $\boldsymbol{X}_i \subseteq \mathcal{D} \sim \mathcal{P}$. That is*

$$|\ell_i(a(\boldsymbol{X}_1)) - \ell_i(a'(\boldsymbol{X}_2))| \leq G\|a(\boldsymbol{X}_1) - a'(\boldsymbol{X}_2)\|_2 \quad \text{for any } a, a' \in \mathcal{A}, \boldsymbol{X}_1, \boldsymbol{X}_2 \subseteq \mathcal{D} \sim \mathcal{P} \quad (16)$$

*Proof.* First, let $g : \mathcal{X} \to \mathbb{R}$ be a function satisfying the following property with $c > 0$:

$$|g(\boldsymbol{X}_1, \ldots, \boldsymbol{X}_{i-1}, \boldsymbol{X}_i, \boldsymbol{X}_{i+1}, \ldots, \boldsymbol{X}_N) - g(\boldsymbol{X}_1, \ldots, \boldsymbol{X}_{i-1}, \boldsymbol{X}_i', \boldsymbol{X}_{i+1}, \ldots, \boldsymbol{X}_N)| \leq c \quad (17)$$

for all $\boldsymbol{X}_1, \ldots, \boldsymbol{X}_N, \boldsymbol{X}_i' \in \mathcal{X}$. Then, for all $\epsilon > 0$,

$$\mathbb{P}(|g(\boldsymbol{X}_1, \ldots, \boldsymbol{X}_N) - \mathbb{E}[g(\boldsymbol{X}_1, \ldots, \boldsymbol{X}_N)]| \geq \epsilon) \leq \exp\left(-\frac{2\epsilon^2}{Nc^2}\right) \quad (18)$$

Take $g$ to be $g = \sup_{\|\boldsymbol{\delta}\| \leq D}(\hat{\mathcal{L}}(\boldsymbol{\delta}_0) - \hat{\mathcal{L}}(\boldsymbol{\delta}) - \mathcal{L}(\boldsymbol{\delta}_0) + \mathcal{L}(\boldsymbol{\delta}))$, which is a function of $\boldsymbol{X}_1, \ldots, \boldsymbol{X}_N$. Since $\|\boldsymbol{\delta}\|_* \leq D$ implies $\|\boldsymbol{\delta}\|_{\mathrm{F}} \leq D$ and by the Lipschitz continuity of $\ell(\cdot, \boldsymbol{Y}_i)$, we have the following for any $(\boldsymbol{X}_i, \boldsymbol{Y}_i) \in \mathcal{D}$:

$$|\ell(f_{\boldsymbol{W}_0}(\boldsymbol{X}_i) + \langle \boldsymbol{G}(\boldsymbol{X}_i), \boldsymbol{\delta}_0 \rangle, \boldsymbol{Y}_i) - \ell(f_{\boldsymbol{W}_0}\boldsymbol{X}_i) + \langle \boldsymbol{G}(\boldsymbol{X}_i), \boldsymbol{\delta} \rangle, \boldsymbol{Y}_i)| \leq G\|\langle \boldsymbol{\delta}_0 - \boldsymbol{\delta}, \boldsymbol{G}(\boldsymbol{X}_i) \rangle\|_2$$

$$\leq G\sqrt{\sum_{j=1}^{K} \|\boldsymbol{\delta}_0 - \boldsymbol{\delta}\|_{\mathrm{F}}^2 \|\boldsymbol{G}^{(j)}(\boldsymbol{X}_i)\|_{\mathrm{F}}^2}$$

$$\leq G\sqrt{\sum_{j=1}^{K} \|\boldsymbol{\delta}_0 - \boldsymbol{\delta}\|_*^2 \|\boldsymbol{G}^{(j)}(\boldsymbol{X}_i)\|_F^2}$$

$$\leq G\sqrt{\sum_{j=1}^{K} 4D^2 \cdot R^2}$$

$$= 2GRD\sqrt{K}. \tag{19}$$

$\square$

Thus, from this lemma, we apply it to LoRA-based continual learning and have

$$\mathcal{L}_i(\boldsymbol{W}_0 + \boldsymbol{B}_t\boldsymbol{A}_t) - \mathcal{L}_i(\boldsymbol{W}_0 + \boldsymbol{B}_i\boldsymbol{A}_i) \leq G\|\langle \boldsymbol{B}_t\boldsymbol{A}_t - \boldsymbol{B}_i\boldsymbol{A}_i, \nabla_{\boldsymbol{W}} f_{\boldsymbol{W}_0}(\boldsymbol{X}_i)\rangle\|_2$$

$$\leq G\sqrt{\sum_{j=1}^{K} \|\boldsymbol{B}_t\boldsymbol{A}_t - \boldsymbol{B}_i\boldsymbol{A}_i\|_F^2 \|\nabla_{\boldsymbol{W}} f_{\boldsymbol{W}_0}^{(j)}(\boldsymbol{X}_i)\|_F^2}$$

$$\leq G\sqrt{\sum_{j=1}^{K} \|\boldsymbol{B}_t\boldsymbol{A}_t - \boldsymbol{B}_i\boldsymbol{A}_i\|_F^2 R^2} \tag{20}$$

where $K$ is the output dimension. From this, we can see that to minimize forgetting error, we should make $\|\boldsymbol{B}_t\boldsymbol{A}_t - \boldsymbol{B}_i\boldsymbol{A}_i\|_F$ as small as possible. Similarly, to minimize intransigence error, $\|\boldsymbol{B}_i\boldsymbol{A}_i - \boldsymbol{B}_i^*\boldsymbol{A}_i^*\|_F$ should be also small. Besides, it's evident that $\mathcal{L}_t(\boldsymbol{W}_0 + \boldsymbol{B}_t\boldsymbol{A}_t) - \mathcal{L}_t(\boldsymbol{W}_0 + \boldsymbol{B}_t\boldsymbol{A}_t) = 0$. Thus, if we makes $\|\boldsymbol{B}_t\boldsymbol{A}_t - \boldsymbol{B}_i\boldsymbol{A}_i\|_F \leq D$ and $\|\boldsymbol{B}_i\boldsymbol{A}_i - \boldsymbol{B}_i^*\boldsymbol{A}_i^*\|_F \leq D$, then we have

$$\mathcal{F}_t + \mathcal{I}_t$$

$$= \sum_{i=1}^{t-1} \left( \mathcal{L}_i(\boldsymbol{W}_0 + \boldsymbol{B}_t\boldsymbol{A}_t) - \mathcal{L}_i(\boldsymbol{W}_0 + \boldsymbol{B}_i\boldsymbol{A}_i) \right) + \sum_{i=1}^{t} \left( \mathcal{L}_i(\boldsymbol{W}_0 + \boldsymbol{B}_i\boldsymbol{A}_i) - \mathcal{L}_i(\boldsymbol{W}_0 + \boldsymbol{B}_i^*\boldsymbol{A}_i^*) \right)$$

$$= \sum_{i=1}^{t} \left( \mathcal{L}_i(\boldsymbol{W}_0 + \boldsymbol{B}_t\boldsymbol{A}_t) - \mathcal{L}_i(\boldsymbol{W}_0 + \boldsymbol{B}_i\boldsymbol{A}_i) \right) + \sum_{i=1}^{t} \left( \mathcal{L}_i(\boldsymbol{W}_0 + \boldsymbol{B}_i\boldsymbol{A}_i) - \mathcal{L}_i(\boldsymbol{W}_0 + \boldsymbol{B}_i^*\boldsymbol{A}_i^*) \right)$$

$$= \sum_{i=1}^{t} \left( \mathcal{L}_i(\boldsymbol{W}_0 + \boldsymbol{B}_t\boldsymbol{A}_t) - \mathcal{L}_i(\boldsymbol{W}_0 + \boldsymbol{B}_i\boldsymbol{A}_i) \right) + \left( \mathcal{L}_i(\boldsymbol{W}_0 + \boldsymbol{B}_i\boldsymbol{A}_i) - \mathcal{L}_i(\boldsymbol{W}_0 + \boldsymbol{B}_i^*\boldsymbol{A}_i^*) \right)$$

$$\leq \sum_{i=1}^{t} G\sqrt{\sum_{j=1}^{K} \|\boldsymbol{B}_t\boldsymbol{A}_t - \boldsymbol{B}_i\boldsymbol{A}_i\|_F^2 R^2} + G\sqrt{\sum_{j=1}^{K} \|\boldsymbol{B}_i\boldsymbol{A}_i - \boldsymbol{B}_i^*\boldsymbol{A}_i^*\|_F^2 R^2}$$

$$= GR\sum_{i=1}^{t} \sqrt{\sum_{j=1}^{K} \|\boldsymbol{B}_t\boldsymbol{A}_t - \boldsymbol{B}_i\boldsymbol{A}_i\|_F^2} + \sqrt{\sum_{j=1}^{K} \|\boldsymbol{B}_i\boldsymbol{A}_i - \boldsymbol{B}_i^*\boldsymbol{A}_i^*\|_F^2}$$

$$\leq 4DGR\sum_{i=1}^{t} \sqrt{K} \tag{21}$$

To make both $\|\boldsymbol{B}_t\boldsymbol{A}_t - \boldsymbol{B}_i\boldsymbol{A}_i\|_F$ and $\|\boldsymbol{B}_i\boldsymbol{A}_i - \boldsymbol{B}_i^*\boldsymbol{A}_i^*\|_F$ minimized, the forgetting-intransigence decomposition can be written as:

$$\|\boldsymbol{B}_t\boldsymbol{A}_t - \boldsymbol{B}_i\boldsymbol{A}_i\|_F + \|\boldsymbol{B}_i\boldsymbol{A}_i - \boldsymbol{B}_i^*\boldsymbol{A}_i^*\|_F$$
$$\leq \|\boldsymbol{B}_t(\boldsymbol{A}_t - \boldsymbol{A}_i)\|_F + \|(\boldsymbol{B}_t - \boldsymbol{B}_i)\boldsymbol{A}_i\|_F + \|\boldsymbol{B}_i(\boldsymbol{A}_i - \boldsymbol{A}_i^*)\|_F + \|(\boldsymbol{B}_i - \boldsymbol{B}_i^*)\boldsymbol{A}_i^*\|_F \tag{22}$$

**Algorithmic motivation.** From the above bound, we observe that generalization error in CL with LoRA depends asymmetrically on the choice of frozen and trainable components. Interestingly, this insight contrasts with standard fine-tuning practices. For example, as concluded in (Zhu et al., 2024b), freezing $\boldsymbol{A}$ and fine-tuning $\boldsymbol{B}$ is at least as effective, if not better, than the reverse. However, if we apply freezing $\boldsymbol{A}$ in CL, it implies $\|\boldsymbol{A}_t - \boldsymbol{A}_i\|_F = 0$, which may unintentionally increase $\|\boldsymbol{A}_i - \boldsymbol{A}_i^*\|_F$ due to limited task-specific expressiveness. In Hu et al. (2022), $\boldsymbol{A}$ is initialized to a random Gaussian matrix satisfying $\mathbb{E}[\boldsymbol{A}^{(0)}(\boldsymbol{A}^{(0)})^\top] = \boldsymbol{I}_r$. Instead, if we propose to fine-tune $\boldsymbol{A}_i$ while extracting orthogonal basis $\boldsymbol{Q}_{i-1}$ from $\boldsymbol{A}_{i-1}$, where $\boldsymbol{Q}_{i-1}\boldsymbol{Q}_{i-1}^\top = \boldsymbol{I}_r$, to initialize $\boldsymbol{A}_i^{(0)}$ via $\boldsymbol{Q}_{i-1}$, then we will have $\boldsymbol{A}_i^{(0)}(\boldsymbol{A}_i^{(0)})^\top = \boldsymbol{I}_r$ where $i \in [1, \ldots, t]$. This orthonormal structure not

only keeps geometric consistency across tasks but also allows $A_j$ $(t \geq j > i)$, to remain well-aligned with previous $A_i$, i.e. $\mathbb{E}[A_j A_i^\top] \approx I_r$, thereby minimizing both $\|A_t - A_i\|_F$ and $\|A_i - A_i^*\|_F$. This motivates our design of continual merging with orthogonal initialization to reduce forgetting and maintain adaptability.

## B.2 Dynamics of low-rank adapters updated by SLAO

We analyze the dynamics of $A_i^s$ and $B_i^s$ in continual learning setting. This analysis, under the orthogonal initialization of $A$, suggests that $B$ may update across different initialization subspaces, effectively increasing the rank of $B$ and thereby aiding generalization.

**Theorem 2.** *Let the parameters $A$ and $B$ be updated using SGD at each step $s$ for task $i$ as follows:*

$$A_i^{s+1} = A_i^s - \eta(B_i^s)^\top(\nabla_W \mathcal{L}_i^s), \quad B_i^{s+1} = B_i^s - \eta(\nabla_W \mathcal{L}_i^s)(A_i^s)^\top \tag{23}$$

*where $\eta$ is the learning rate. We assume $A_i^s = A_i^{(0)} + \eta A_i^{(0)} f_A(s)$ and $B_i^s = B_i^{(0)} + \eta f_B(s)(A_i^{(0)})^\top$ holds with such functions $f_A$ and $f_B$ for $1, \ldots, s$, and $\|\sum_{s=1}^S \nabla_W \mathcal{L}_i^{(s)}\|_F \leq L$ for every $S$ during training task $i$, which implies that the model stays within a finite Euclidean ball. If we assume $A_i^{(0)}(A_i^{(0)})^\top = I_r$, in this case, the dynamics of $A_i$ satisfies $\|f_A(s)\|_2 \leq \frac{\eta L^2(1-(\eta^2 L^2)^s)}{1-\eta^2 L^2}$, and the dynamics of $B$ satisfies $f_B(s) = -\sum_{j=0}^{s-1}(\nabla_W \mathcal{L}_i^j)(\eta f_A^\top(j) + I)$. When $\eta$ is small, we have $f_B(s) \approx -\sum_{j=0}^{s-1}(\nabla_W \mathcal{L}_i^j)$. Thus $B_i^S = \eta f_B(S)(A_i^{(0)})^\top$, and total update for $B_i$ is $\Delta B_i = -\eta\left(\sum_{s=0}^S(\nabla_W \mathcal{L}_i^s)\right)(A_i^{(0)})^\top$.*

*Proof.* We start by noting the fact that for **Task 1**, when $s = 0$, $f_A(0) = f_B(0) = 0$. For $s > 0$, assume $A_1^s = A_0 + \eta A_0 f_A(s)$ and $B_1^s = \eta f_B(s) A_0^\top$. Since the first task training is the same as the LoRA fine-tuning in Hao et al. (2024) for the dynamics of $A$ and $B$ we have:

$$\begin{aligned} A_1^{s+1} &= A_1^s - \eta(B_1^s)^\top(\nabla_{W_0} \mathcal{L}_1^s) \\ &= A_0 + \eta A_0 f_A(s) - \eta^2 A_0 f_B^\top(s)(\nabla_{W_0} \mathcal{L}_1^s) \\ &= A_0 + \eta A_0 f_A(s+1) \end{aligned} \tag{24}$$

and

$$\begin{aligned} B_1^{s+1} &= B_1^s - \eta(\nabla_{W_0} \mathcal{L}_1^s)(A_1^s)^\top \\ &= \eta f_B(s+1) A_0^\top \end{aligned} \tag{25}$$

Thus, by rearranging the terms, we have:

$$f_A(s) = -\eta \sum_{j=0}^{s-1} f_B^\top(j)(\nabla_{W_0} \mathcal{L}_1^j) \tag{26}$$

$$f_B(s) = -\sum_{j=0}^{s-1}(\nabla_{W_0} \mathcal{L}_1^j)(\eta f_A^\top(j) + I) \tag{27}$$

Since $W_0 + BA \approx W_0 + \Delta B A_0$ when learning rate $\eta$ is small, the change in $B$ dominates the final weight update. Thus, if freezing $A$, we obtain

$$\Delta B_1 \approx -\eta\left(\sum_{s=1}^S \nabla_{W_0} \mathcal{L}_1^s\right) A_0^\top \tag{28}$$

and

$$f_B(s) \approx -\sum_{j=0}^{s-1} \nabla_{W_0} \mathcal{L}_1^j \tag{29}$$

**Task $i$ ($i > 1$):** when $s = 0$, $f_A(0) = f_B(0) = 0$.

For $s > 0$: Assume $A_i^s = A_i^{(0)} + \eta A_i^{(0)} f_A(s)$ and $B_i^s = B_i^{(0)} + \eta f_B(s)(A_i^{(0)})^\top$ hold with such functions $f_A$ and $f_B$ for $1, \ldots, s$. Then, for $s + 1$, we have

$$A_i^{s+1} = A_i^s - \eta(B_i^s)^\top(\nabla_{W_0} \mathcal{L}_i^s)$$

$$\begin{aligned}
&= \boldsymbol{A}_i^{(0)} + \eta \boldsymbol{A}_i^{(0)} f_A(s) - \eta(\boldsymbol{B}_i^{(0)} + \eta f_B(s)(\boldsymbol{A}_i^{(0)})^\top)^\top (\nabla_{\boldsymbol{W}_0} \mathcal{L}_i^s) \\
&= \boldsymbol{A}_i^{(0)} + \eta \boldsymbol{A}_i^{(0)} f_A(s) - \eta((\boldsymbol{B}_i^{(0)})^\top + \eta \boldsymbol{A}_i^{(0)} f_B^\top(s))(\nabla_{\boldsymbol{W}_0} \mathcal{L}_i^s) \\
&= \boldsymbol{A}_i^{(0)} + \eta \boldsymbol{A}_i^{(0)}(f_A(s) - \eta f_B^\top(s)(\nabla_{\boldsymbol{W}_0} \mathcal{L}_i^s)) - \eta(\boldsymbol{B}_i^{(0)})^\top(\nabla_{\boldsymbol{W}_0} \mathcal{L}_i^s)
\end{aligned} \tag{30}$$

We would like to express $\boldsymbol{A}_i^{s+1}$ as

$$\boldsymbol{A}_i^{s+1} = \boldsymbol{A}_i^{(0)} + \eta \boldsymbol{A}_i^{(0)} f_A(s+1) \tag{31}$$

So compare both sides:

$$\eta \boldsymbol{A}_i^{(0)} f_A(s+1) = \eta \boldsymbol{A}_i^{(0)}(f_A(s) - \eta f_B^\top(s)(\nabla_{\boldsymbol{W}_0} \mathcal{L}_i^s)) - \eta(\boldsymbol{B}_i^{(0)})^\top(\nabla_{\boldsymbol{W}_0} \mathcal{L}_i^s) \tag{32}$$

Divide both sides by $\eta$, rearrange:

$$\boldsymbol{A}_i^{(0)} f_A(s+1) = \boldsymbol{A}_i^{(0)}(f_A(s) - \eta f_B^\top(s)(\nabla_{\boldsymbol{W}_0} \mathcal{L}_i^s)) - (\boldsymbol{B}_i^{(0)})^\top(\nabla_{\boldsymbol{W}_0} \mathcal{L}_i^s) \tag{33}$$

Since by our initialization $\boldsymbol{A}_i^{(0)}(\boldsymbol{A}_i^{(0)})^\top = \boldsymbol{I}_r$, then we have

$$\boldsymbol{A}_i^{(0)} f_A(s+1) = \boldsymbol{A}_i^{(0)}(f_A(s) - \eta f_B^\top(s)(\nabla_{\boldsymbol{W}_0} \mathcal{L}_i^s)) - \boldsymbol{I}_r(\boldsymbol{B}_i^{(0)})^\top(\nabla_{\boldsymbol{W}_0} \mathcal{L}_i^s) \tag{34}$$

$$\boldsymbol{A}_i^{(0)} f_A(s+1) = \boldsymbol{A}_i^{(0)}(f_A(s) - \eta f_B^\top(s)(\nabla_{\boldsymbol{W}_0} \mathcal{L}_i^s) - (\boldsymbol{A}_i^{(0)})(\boldsymbol{A}_i^{(0)})^\top(\boldsymbol{B}_i^{(0)})^\top(\nabla_{\boldsymbol{W}_0} \mathcal{L}_i^s) \tag{35}$$

$$f_A(s+1) = f_A(s) - \eta f_B^\top(s)(\nabla_{\boldsymbol{W}_0} \mathcal{L}_i^s) - (\boldsymbol{A}_i^{(0)})^\top(\boldsymbol{B}_i^{(0)})^\top(\nabla_{\boldsymbol{W}_0} \mathcal{L}_i^s) \tag{36}$$

$$f_A(s+1) = -\eta \sum_{j=0}^{s} \left( f_B^\top(j) - (\boldsymbol{A}_i^{(0)})^\top(\boldsymbol{B}_i^{(0)})^\top \right)(\nabla_{\boldsymbol{W}_0} \mathcal{L}_i^j) \tag{37}$$

For $\boldsymbol{B}$, we have:

$$\begin{aligned}
\boldsymbol{B}_i^{s+1} &= \boldsymbol{B}_i^s - \eta(\nabla_{\boldsymbol{W}_0} \mathcal{L}_i^s)(\boldsymbol{A}_i^s)^\top \\
&= \boldsymbol{B}_i^{(0)} + \eta f_B(s)(\boldsymbol{A}_i^{(0)})^\top - \eta(\nabla_{\boldsymbol{W}_0} \mathcal{L}_i^s)(\boldsymbol{A}_i^{(0)} + \eta \boldsymbol{A}_i^{(0)} f_A(s))^\top \\
&= \boldsymbol{B}_i^{(0)} + \eta(f_B(s)(\boldsymbol{A}_i^{(0)})^\top - (\nabla_{\boldsymbol{W}_0} \mathcal{L}_i^s)(\boldsymbol{A}_i^{(0)} + \eta \boldsymbol{A}_i^{(0)} f_A(s))^\top) \\
&= \boldsymbol{B}_i^{(0)} + \eta(f_B(s)(\boldsymbol{A}_i^{(0)})^\top - (\nabla_{\boldsymbol{W}_0} \mathcal{L}_i^s)(\boldsymbol{A}_i^{(0)})^\top - \eta(\nabla_{\boldsymbol{W}_0} \mathcal{L}_i^s) f_A^\top(s)(\boldsymbol{A}_i^{(0)})^\top) \\
&= \boldsymbol{B}_i^{(0)} + \eta \left( f_B(s) - (\nabla_{\boldsymbol{W}_0} \mathcal{L}_i^s) - \eta(\nabla_{\boldsymbol{W}_0} \mathcal{L}_i^s) f_A^\top(s) \right)(\boldsymbol{A}_i^{(0)})^\top
\end{aligned} \tag{38}$$

Thus,

$$\begin{aligned}
f_B(s+1) &= f_B(s) - (\nabla_{\boldsymbol{W}_0} \mathcal{L}_i^s) - \eta(\nabla_{\boldsymbol{W}_0} \mathcal{L}_i^s) f_A^\top(s) \\
&= f_B(s) - (\nabla_{\boldsymbol{W}_0} \mathcal{L}_i^s)(\eta f_A^\top(s) + \boldsymbol{I}) \\
&= -\sum_{j=0}^{s} (\nabla_{\boldsymbol{W}_0} \mathcal{L}_i^j)(\eta f_A^\top(j) + \boldsymbol{I})
\end{aligned} \tag{39}$$

Then, we have:

$$\begin{aligned}
\|f_A(s)\|_{\mathrm{F}} &= \left\| \eta \sum_{j=0}^{s-1} \left( \sum_{m=0}^{j-1} (\eta f_A(m) + \boldsymbol{I})(\nabla_{\boldsymbol{W}_0} \mathcal{L}_i^m)^\top + (\boldsymbol{A}_i^{(0)})^\top(\boldsymbol{B}_i^{(0)})^\top \right)(\nabla_{\boldsymbol{W}_0} \mathcal{L}_i^j) \right\|_{\mathrm{F}} \\
&\leq \eta^2 \left\| \sum_{m=0}^{s-2} (f_A(j))(\nabla_{\boldsymbol{W}_0} \mathcal{L}_i^m)^\top \sum_{j=m+1}^{s-1} (\nabla_{\boldsymbol{W}_0} \mathcal{L}_i^j) \right\|_{\mathrm{F}} \\
&\quad + \eta \left\| \sum_{j=0}^{s-1} \sum_{m=0}^{j-1} (\nabla_{\boldsymbol{W}_0} \mathcal{L}_i^m)^\top(\nabla_{\boldsymbol{W}_0} \mathcal{L}_i^j) \right\|_{\mathrm{F}} \\
&\quad + \eta \left\| \sum_{j=0}^{s-1} (\boldsymbol{A}_i^{(0)})^\top(\boldsymbol{B}_i^{(0)})^\top(\nabla_{\boldsymbol{W}_0} \mathcal{L}_i^j) \right\|_{\mathrm{F}}
\end{aligned}$$

$$\leq \eta^2 L \left\| \sum_{m=0}^{s-2} (f_A(j))(\nabla_{\boldsymbol{W}_0}\mathcal{L}_i^m)^\top \right\|_{\mathrm{F}} + \eta L^2 + \eta^2 L^2 s \|(\boldsymbol{A}_i^{(0)})^\top (\boldsymbol{B}_i^{(0)})^\top\|_{\mathrm{F}} \tag{40}$$

$$\|(\boldsymbol{A}_i^{(0)})^\top (\boldsymbol{B}_i^{(0)})^\top\|_{\mathrm{F}} = \|(\boldsymbol{B}_i^{(0)} \boldsymbol{A}_i^{(0)})^\top\|_{\mathrm{F}} = \sqrt{\sum_{p=1}^{r} \sigma_p^2(\boldsymbol{B}_i^{(0)} \boldsymbol{A}_i^{(0)})} \leq \sqrt{r} \tag{41}$$

If $\|f_A(s)\| \leq a_s = \frac{\eta L^2(1-(\eta^2 L^2)^s)}{1-\eta^2 L^2}$, then

$$
\begin{aligned}
\|f_A(s)\|_{\mathrm{F}} &\leq \eta^2 L^2 a_{s-1} + \eta L^2 + \eta^2 L^2 s \sqrt{r} \\
&= \eta^2 L^2 \frac{\eta L^2(1-(\eta^2 L^2)^s)}{1-\eta^2 L^2} + \eta L^2 + \eta^2 L^2 s\sqrt{r} + \eta^2 L^2 s\sqrt{r} \\
&= \frac{\eta^3 L^4(1-(\eta^2 L^2)^s) + \eta L^2 - \eta^3 L^4 + \eta^2 L^2 s\sqrt{r} - \eta^4 L^4 s\sqrt{r}}{1-\eta^2 L^2} \\
&= \frac{\eta L^2(1-(\eta^2 L^2)^s) + \eta^2 L^2 s\sqrt{r}(1-\eta^2 L^2)}{1-\eta^2 L^2}
\end{aligned}
\tag{42}
$$

We have

$$\|f_A(s)\|_2 \leq \|f_A(s)\|_{\mathrm{F}} \tag{43}$$

If $\eta \ll 1/L$,

$$
\begin{aligned}
\eta\|f_A(s)\|_{\mathrm{F}} &\leq \eta \frac{\eta L^2(1-(\eta^2 L^2)^s) + \eta^2 L^2 s\sqrt{r}(1-\eta^2 L^2)}{1-\eta^2 L^2} \\
&\leq \eta \frac{\eta L^2(1-(\eta^2 L^2)^s)}{1-\eta^2 L^2} \\
&\leq \eta a_s
\end{aligned}
\tag{44}
$$

Thus, we have $\eta\|f_A(s)\| \leq \eta a_s$. The dynamics are:

$$f_A(s) = -\eta \sum_{j=0}^{s-1} f_B^\top(j)(\nabla_{\boldsymbol{W}_0}\mathcal{L}_i^j), \quad f_B(s) = -\sum_{j=0}^{s-1}(\nabla_{\boldsymbol{W}_0}\mathcal{L}_i^j)(\eta f_A^\top(j) + \boldsymbol{I}) \tag{45}$$

In our algorithm, we fine-tune $\boldsymbol{A}_i^{(0)}$ in our algorithm and we have $\eta\|f_A(s)\| \ll \boldsymbol{I}$, then

$$f_A(s) = -\eta \sum_{j=0}^{s-1}\left(-\sum_{m=0}^{j-1}(\nabla_{\boldsymbol{W}_0}\mathcal{L}_i^m)\right)^\top (\nabla_{\boldsymbol{W}_0}\mathcal{L}_i^j), \quad f_B(s) = -\sum_{j=0}^{s-1}(\nabla_{\boldsymbol{W}_0}\mathcal{L}_i^j) \tag{46}$$

Therefore, we have

$$\Delta \boldsymbol{B}_i \approx -\eta \left(\sum_{s=0}^{S} \nabla_{\boldsymbol{W}_0}\mathcal{L}_i^s\right)(\boldsymbol{A}_i^{(0)})^\top \tag{47}$$

$\square$

## C   OVERVIEW OF SLAO

We show the detailed overview of SLAO in Figure 4. It presents a framework where fine-tuned LORA and merged LoRA are processed over time, and specializes key components: (1) orthogonal initialization, and (2) time-aware continual merging.

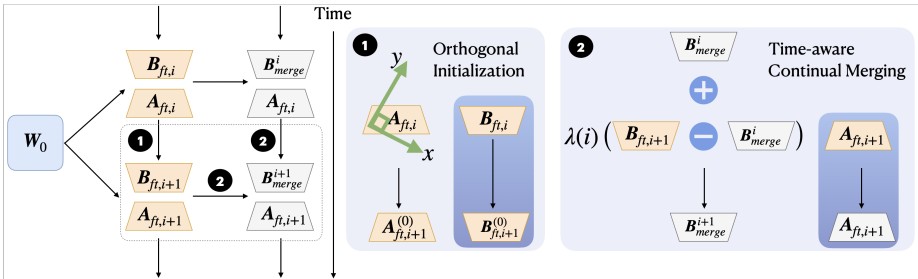

Figure 4: Overview of SLAO. Left area is a framework where fine-tuned LoRA (orange) and merged LoRA (gray) are processed over time. Right area (2 blue boxes) highlights key components: (1) Orthogonal initialization for new task $i+1$ learning LoRA, where orthogonal basis is extracted from $A_{\text{ft},i}$ to initialize $A_{\text{ft},i+1}$ such that $A_{\text{ft},i+1}^{(0)}(A_{\text{ft},i+1}^{(0)})^\top = I_r$ and $B_{\text{ft},i+1}$ is initialized by previous $B_{\text{ft},i}$; (2) Time-aware continual merging for $B_{\text{ft},i+1}$ and $B_{\text{merge}}^i$, and update $A_{\text{merge}}^{i+1}$ via $A_{\text{ft},i+1}$.

## D    EXPERIMENTS DETAILS

Our experiments are conducted on 4 NVIDIA A100 GPUs using the DeepSpeed repository. We evaluate five LLMs: Llama-2-7B-chat, Llama-2-13B-chat, Llama-3-2-3B, Qwen2.5-3B, and Qwen2.5-7B. Each individual experiment (e.g., running a single task order from the large number of tasks benchmark on Llama-2-13B-chat) can be executed on a single A100 GPU. We apply LoRA to the query and value projection matrices in the attention modules of each model, with a fixed rank of 8. Each task order is evaluated over 3 random seeds.

For Standard CL benchmark and the large number of tasks benchmark on Llama-2-7B-chat and Llama-2-13B-chat, we follow the setting in Wang et al. (2023), and train the models with one epoch, a constant learning rate of 1e-4, the training batch size is 1, and the gradient accumulation step is 8. And for Standard CL benchmark and the large number of tasks benchmark on Llama-3-2-3B, we set the learning rate as 1e-4. For SuperNI benchmark using Llama-2-7B-chat and Llama-2-13B-chat, we follow the learning rate, training batch size, and gradient accumulation steps in Zhao et al. (2024b), and we use 5e-5 and train the models with five epochs with a training batch size of 2, and gradient accumulation steps of 4. And for SuperNI benchmark on Llama-3-2-3B, we set the learning rate as 5e-5.

### D.1    OVERALL RESULTS ON LLAMA-2-13B-CHAT

**Continual learning performance analysis.**    As shown in Table 5, our method consistently outperforms all data-free baselines across two benchmarks using Llama-2-13B-chat model.

*LoRA-Based continual learning*: IncLoRA improves upon SeqLoRA by freezing previously learned LoRAs to isolate subspaces, though its subspace separation is simplistic. SeqLoRA performs a little better than O-LoRA in the standard CL benchmark, since we use the hyperparameter $\lambda = 0.5$ in the orthogonal loss in O-LoRA that may be adjusted along with the different models and datasets, while O-LoRA outperforms SeqLoRA and IncLoRA in large number of tasks benchmark. SAPT-LoRA achieves the highest average performance among LoRA-based methods, but it relies on generated previous task pseudo samples, unrealistic in many LLM scenarios, and is more sensitive to task ordering than our method. LoRM-BA (from second task, begin with freezing $B$) and LoRM-AB (from second task, begin with freezing $A$) yield nearly identical results in large number of tasks benchmark, suggesting that the order of alternating LoRA components in learning sequential tasks does not significantly affect outcomes, but LoRM-BA outperforms LoRM-AB in standard CL benchmark. CorDA performs well in standard CL benchmark and large number of tasks, but both of them are lower than our method. MagMax performs comparably to our approach on standard CL benchmark, and slightly worse on large number of tasks, thus only keeping the weights which have the largest absolute value would cause catastrophic forgetting.

*LoRA merging baselines:* KnOTS and LoRA-LEGO perform similarly in the standard CL benchmark, but KnOTS outperforms in the large number of tasks. KnOTS may benefit from flexible SVD-merging mechanism so that we apply time-aware scaling on merging, while LoRA-LEGO treats tasks equally, lacks prioritization, and is ineffective in complex CL contexts.

Table 5: Testing performance (%) on three CL benchmarks using Llama-2-13B-chat across different task orders, where each result is run three random times, where $Oi$ denotes $i$th task order.

| Method | Standard CL Benchmark | | | | Large Number of Tasks | | | |
|---|---|---|---|---|---|---|---|---|
| | O1 | O2 | O3 | avg | O4 | O5 | O6 | avg |
| SeqLoRA | 77.1 | 77.4 | 78.6 | 77.7 | 74.1 | 74.2 | 74.5 | 74.3 |
| IncLoRA | 78.3 | 79.6 | 79.5 | 79.1 | 74.2 | 76.1 | 75.1 | 75.1 |
| O-LoRA | 76.1 | 77.1 | 78.5 | 77.2 | 75.5 | 75.5 | 74.8 | 75.3 |
| SPAT-LoRA | 83.2 | 82.4 | 80.1 | 81.9 | 83.9 | 80.2 | 82.3 | 82.1 |
| LoRM-BA | 78.4 | 80.2 | 80.4 | 79.7 | 74.9 | 71.9 | 70.5 | 72.4 |
| LoRM-AB | 76.2 | 74.1 | 75.3 | 75.2 | 74.3 | 70.1 | 72.3 | 72.2 |
| CorDA | 79.1 | 80.4 | 80.6 | 80.0 | 75.9 | 75.8 | 72.9 | 74.9 |
| MagMax | 80.9 | 80.6 | 80.7 | 80.7 | 73.7 | 73.4 | 76.0 | 74.4 |
| KnOTS (zero init) | 71.9 | 73.1 | 73.9 | 73.0 | 66.8 | 65.5 | 66.0 | 66.1 |
| LoRA-LEGO | 73.0 | 72.3 | 72.9 | 72.7 | 64.3 | 63.3 | 64.0 | 63.9 |
| OPCM | 68.8 | 63.6 | 61.6 | 64.7 | 57.6 | 60.2 | 58.8 | 58.9 |
| SLAO (ours) | 80.8 | 81.1 | 81.1 | 81.0 | 76.5 | 75.9 | 76.1 | 76.2 |
| Multi-Task | 81.4 | | | | 79.2 | | | |

Table 6: Comparison of merging strategies on testing performance on two CL benchmarks using Llama-2-13B-chat across different task orders, where $Oi$ denotes $i$th task order.

| Method | Standard CL Benchmark | | | | Large Number of Tasks | | | |
|---|---|---|---|---|---|---|---|---|
| | O1 | O2 | O3 | avg | O4 | O5 | O6 | avg |
| FREB-MA | 77.7 | 77.5 | 76.4 | 77.2 | 69.1 | 68.2 | 69.3 | 68.9 |
| FREA-MB | 79.4 | 79.2 | 78.9 | 79.2 | 73.2 | 73.8 | 73.2 | 73.4 |
| FTBA-MA | 79.2 | 80.1 | 80.8 | 80.0 | 75.1 | 75.0 | 75.8 | 75.3 |
| FTBA-MBA | 80.7 | 80.9 | 80.8 | 80.8 | 75.4 | 75.4 | 76.0 | 75.6 |
| FTBA-MB | 80.6 | 80.9 | 80.9 | 80.8 | 75.8 | 75.7 | 76.0 | 75.8 |
| SLAO (ours) | 80.8 | 81.1 | 81.1 | 81.0 | 76.5 | 75.9 | 76.1 | 76.2 |

***Continual merging approaches:*** since OPCM is designed for full model, directly applying it to LoRA by treating its two components identically leads to suboptimal performance.

**Asymmetry in LoRA merging.** To investigate the asymmetry in LoRA merging, we compare different continual merging strategies for LoRA: (1) Freeze A, merge B (FREA-MB), (2) Freeze B, merge A (FREB-MA), (3) Fine-tune BA, merge A (FTBA-MA), (4) Fine-tune BA, merge BA (FTBA-MBA), (5) Fine-tune BA, merge B (FTBA-MB). As shown in Table 6, only fine-tuning LoRA and merging $B$ consistently outperforms other strategies, except ours. Fine-tuning LoRA and merging $BA$ has comparable performance compared to fine-tuning LoRA and merging $B$, but fine-tuning LoRA and merging $A$ yields the poorest performance among fine-tuning LoRA methods. When freezing one component of LoRA during training, freezing $A$ and merging $B$ is much better than freezing $B$ and merging $A$, consistent with conclusion in Zhu et al. (2024b) that freezing $A$ and fine-tuning $B$ is at least better than freezing $B$ and fine-tuning $A$. This highlights the asymmetry in LoRA components and the importance of asymmetric merging based on their fundamental roles in adaptation.

## D.2 RESULTS IN QWEN2.5 MODELS

To evaluate the performance of our SLAO using state-of-the-art LLMs, we use Qwen2.5-3B and Qwen2.5-7B to compare with the other two LoRA-based continual learning baselines on the SuperNI benchmark.

Table 7: Comparison of testing performance on SuperNI benchmark using Qwen2.5-3B and Qwen2.5-7B models across two different task orders.

| Method | Qwen2.5-3B | | | Qwen2.5-7B | | |
| --- | --- | --- | --- | --- | --- | --- |
| | O1 | O2 | avg | O1 | O2 | avg |
| O-LoRA | 31.1 | 29.8 | 30.5 | 34.3 | 32.6 | 33.4 |
| InfLoRA | 35.6 | 25.6 | 30.6 | 43.5 | 31.0 | 37.3 |
| SLAO | 37.8 | 32.4 | 35.1 | 41.0 | 35.5 | 38.3 |

Table 8: Comparison of initialization strategies for existing LoRA merging methods on testing performance on two CL benchmarks using Llama-2-7B-chat and Llama-2-13B-chat across different task orders, where $Oi$ denotes $i$th task order.

| Init | Model | Method | Standard CL Benchmark | | | | Large Number of Tasks | | | |
| --- | --- | --- | --- | --- | --- | --- | --- | --- | --- | --- |
| | | | O1 | O2 | O3 | avg | O4 | O5 | O6 | avg |
| Random (Zero) | 7B | KnOTS | 67.9 | 65.9 | 70.8 | 68.2 | 61.5 | 60.1 | 58.0 | 59.9 |
| Random (Zero) | 7B | LoRA-LEGO | 68.3 | 66.0 | 70.9 | 68.4 | 58.8 | 58.7 | 53.2 | 56.9 |
| Last-FT | 7B | KnOTS | 79.7 | 80.7 | 79.9 | 80.1 | 74.0 | 72.5 | 75.0 | 73.8 |
| Last-FT | 7B | LoRA-LEGO | 78.8 | 79.8 | 79.5 | 79.4 | 72.2 | 70.4 | 74.2 | 72.3 |
| Random (Zero) | 13B | KnOTS | 71.9 | 73.1 | 73.9 | 73.0 | 66.8 | 65.5 | 66.0 | 66.1 |
| Random (Zero) | 13B | LoRA-LEGO | 73.0 | 72.3 | 72.9 | 72.7 | 64.3 | 63.3 | 64.0 | 63.9 |
| Last-FT | 13B | KnOTS | 80.1 | 79.6 | 80.3 | 80.0 | 75.4 | 74.9 | 75.8 | 75.4 |
| Last-FT | 13B | LoRA-LEGO | 79.8 | 80.0 | 80.2 | 80.0 | 75.3 | 73.8 | 76.0 | 75.0 |

When evaluating SuperNI benchmarks in Table 7, SLAO is consistently better than other two baselines and achieves the best average performance on both Qwen2.5-3B and Qwen2.5-7B models, demonstrating its robustness on Qwen2.5 model sizes. Also, the performance of SLAO on Qwen2.5-3B is better than that on Qwen2.5-7B.

### D.3 IMPACT OF INITIALIZATION STRATEGIES FOR EXISTING LoRA MERGING METHODS

We compare the testing performance of different initialization strategies for existing LoRA merging methods in Table 8, where we compare two strategies: random (zero) initialization and the last fine-tuning point of previous tasks. For zero initialization for new tasks, when using Llama-2-7B-chat and Llama-2-13B-chat, KnOTS and LoRA-LEGO perform similarly in the standard CL benchmark, but KnOTS outperforms in the large number of tasks. For last fine-tuning point initialization, KnOTS and LoRA-LEGO perform similarly in the standard CL benchmark and the large number of tasks, while KnOTS is slightly over LoRA-LEGO. All results in the last fine-tuning point initialization are significantly better than the zero initialization. These results show that our last fine-tuning point initialization has better performance than zero initialization in continual merging scenarios.

### D.4 CONTINUAL LEARNING PERFORMANCE ON BACKWARD TRANSFER

We evaluate the performance of backward transfer (BWT) using Llama-2-7B-chat on the large number of tasks benchmark. As shown in Table 9, SLAO demonstrates strong backward transfer ability. Among all methods, SeqLoRA performs the worst due to its lack of mechanisms to prevent forgetting. KnOTS and LoRA-LEGO also underperform, as they are primarily designed for model merging rather than continual learning. IncLoRA exhibits limited BWT performance, while O-LoRA achieves better results by enforcing orthogonality during learning. InfLoRA slightly trails O-LoRA, and CorDA performs worse, possibly due to its reliance on nullspace projection without time-aware updates. SAPT-LoRA achieves the best BWT overall, though it benefits from synthetic data from previous tasks, which may not be feasible in realistic settings. Between the two variants of LoRM, LoRM-BA slightly outperforms LoRM-AB. Finally, MagMax and OPCM show comparable performance, both designed to balance update integration during continual merging.

Table 9: Testing performance (%) of the average of backward transfer (BWT) on large number of tasks using Llama-2-7B-chat across different task orders.

| Method | BWT | Method | BWT |
|---|---|---|---|
| SeqLoRA | -17.2 | LoRM-BA | -6.7 |
| IncLoRA | -9.6 | LoRM-AB | -4.1 |
| O-LoRA | -4.0 | MagMax | -3.8 |
| InfLoRA | -4.9 | OPCM | -3.9 |
| SAPT-LoRA | -2.9 | KnOTS (zero init) | -14.1 |
| CorDA | -4.5 | LoRA-LEGO | -15.6 |
| SLAO (ours) | -3.5 | | |

Table 10: Comparison of MOPD and AOPD on testing performance across three standard CL benchmarks using Llama-2-7B-chat in different task orders.

| Method | Standard CL Benchmark | | Large Number of Tasks | | SuperNI Benchmark | |
|---|---|---|---|---|---|---|
| | MOPD | AOPD | MOPD | AOPD | MOPD | AOPD |
| O-LoRA | 9.84% | 5.79% | 17.87% | 8.53% | 22.16% | 11.63% |
| SAPT-LoRA | 8.75% | 5.69% | 18.94% | 9.65% | 25.28% | 12.38% |
| InfLoRA | 8.23% | 2.54% | 19.58% | 10.01% | 23.42% | 11.46% |
| SLAO (ours) | 1.72% | 1.30% | 15.17% | 7.16% | 18.94% | 10.76% |

### D.5 CONTINUAL LEARNING PERFORMANCE ON ORDER-NORMALIZED PERFORMANCE DISPARITY

We evaluate the performance of Order-normalized Performance Disparity (Yoon et al., 2020) using Llama-2-7B-chat on three benchmarks. Order-normalized Performance Disparity is used to evaluate order-sensitivity for each task $t$, defined as the disparity between its performance on $R$ random task sequences:

$$OPD_t = \max(\overline{P}_t^1, \ldots, \overline{P}_t^R) - \min(\overline{P}_t^1, \ldots, \overline{P}_t^R) \tag{48}$$

where $\overline{P}_t^r$ denotes the performance of task $t$ to the task sequence $r$. The Maximum OPD is defined as $MOPD = \max(OPD_1, \ldots, OPD_t)$ and the Average OPD is defined as $AOPD = \frac{1}{T}\sum_{t=1}^{T} OPD_t$, to evaluate order-robustness on the whole task set. Lower scores of both metrics indicate higher robustness.

Table 10 shows the performance of MOPD and AOPD on three benchmarks with their task sequences. Our SLAO shows the most stable performance across different task orders, indicating that it handles order sensitivities better compared to other baselines. While SAPT-LoRA achieves higher scores in Table 1, it heavily depends on past tasks' data information, so that SAPT-LoRA suffers from greater variation in different task orders, mostly due to its past pseudo-sample generation.

### D.6 CHOICE OF ORTHOGONAL DECOMPOSITION STRATEGY

In our algorithm, SLAO, we use QR decomposition to extract the orthogonal basis from previous LoRA $A$. To evaluate the effectiveness of QR decomposition, we compare it against orthogonal bases derived from (1) singular value decomposition (SVD), where the product $UV^\top$ forms an orthogonal approximation, and (2) randomized SVD, where the product $QU$ forms an orthogonal approximation. As shown in Table 11, QR initialization performs similarly to the SVD approach on standard CL benchmark and large number of tasks, but the performance of QR on SuperNI benchmark is better than that of SVD. The performance of randomized SVD is not better than SVD and QR across these three benchmarks.

Table 11: Comparison of orthogonal decomposition strategies on testing performance on two CL benchmarks using Llama-2-7B-chat across different task orders, where $Oi$ denotes $i$th task order.

| Method | Standard CL Benchmark | | | | Large Number of Tasks | | | | SuperNI Benchmark | | |
|---|---|---|---|---|---|---|---|---|---|---|---|
| | O1 | O2 | O3 | avg | O4 | O5 | O6 | avg | O1 | O2 | avg |
| Randomized SVD | 72.6 | 69.1 | 73.3 | 71.7 | 52.3 | 62.5 | 57.8 | 57.5 | 11.8 | 22.8 | 17.3 |
| SVD | 79.9 | 80.8 | 80.2 | 80.3 | 75.3 | 74.4 | 75.1 | 74.9 | 36.9 | 33.7 | 35.3 |
| QR | 80.1 | 80.8 | 80.4 | 80.4 | 75.0 | 74.4 | 75.1 | 74.8 | 38.7 | 35.7 | 37.2 |

### D.7 ASYMMETRY OF LORA

We separately fine-tune 15 tasks from the SuperNI benchmark using 15 LoRAs on Llama-2-7B-chat (Touvron et al., 2023), and compute cosine similarity of $A$ and $B$ across 15 tasks using the last layer LoRA. Figure 5 shows that $A$ exhibits significantly higher similarity across tasks compared to $B$, suggesting that LoRA components follow inherently different learning dynamics.

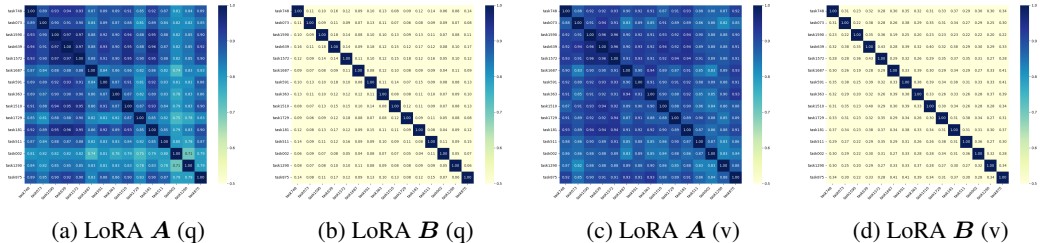

| (a) LoRA $A$ (q) | (b) LoRA $B$ (q) | (c) LoRA $A$ (v) | (d) LoRA $B$ (v) |
|---|---|---|---|

Figure 5: Cosine similarity between 15 tasks from SuperNI benchmark for fine-tuned q and v attention LoRA $A$ and $B$ in the last layer (32nd) of Llama-2-7B-chat.

### D.8 IMPACT OF LEARNING RATE

To assess the effect of different learning rates, we evaluate SeqLoRA on the standard CL benchmark using Llama-2-7B-chat with learning rates of 1e-3, 1e-4, and 1e-5. As shown in Table 12, a learning rate of 1e-4 achieves the best performance, while 1e-3 performs the worst, significantly degrading the overall results. This highlights the importance of careful learning rate selection in LoRA-based continual learning.

Table 12: Impact of learning rate on testing performance of SeqLoRA using Llama-2-7B-chat across three task orders in Standard CL Benchmark, where $Oi$ denotes $i$th task order.

| learning rate | Standard CL Benchmark | | | |
|---|---|---|---|---|
| | O1 | O2 | O3 | avg |
| $1e-3$ | 6.0 | 0.0 | 19.7 | 8.6 |
| $1e-4$ | 73.3 | 76.2 | 78.4 | 76.0 |
| $1e-5$ | 73.6 | 71.8 | 76.0 | 73.8 |

### D.9 IMPACT OF THE RANK OF LORA

We assess the effect of different LoRA rank values in our algorithm by comparing three rank settings on both the standard CL benchmark and the large number of tasks benchmark using Llama-2-13B-chat. As shown in Table 13, a rank of 8 yields the best performance on the standard CL benchmark, while a rank of 4 performs best on the large number of tasks benchmark. Overall, the performance differences across the three ranks are relatively small, suggesting that our method is robust to the choice of rank.

Table 13: Impact of the rank of LoRA on testing performance of SLAO using Llama-2-13B-chat across Standard CL Benchmark and large number of tasks, where $Oi$ denotes $i$th task order.

| rank | Standard CL Benchmark | | | | Large Number of Tasks | | | |
|------|------|------|------|------|------|------|------|------|
| | O1 | O2 | O3 | avg | O4 | O5 | O6 | avg |
| $r = 4$ | 80.6 | 81.0 | 81.1 | 80.9 | 77.2 | 76.3 | 77.1 | 76.9 |
| $r = 8$ | 80.8 | 81.1 | 81.1 | 81.0 | 76.5 | 75.9 | 76.1 | 76.2 |
| $r = 16$ | 80.3 | 80.9 | 80.9 | 80.7 | 76.9 | 75.5 | 75.9 | 76.1 |

Table 14: Comparison of training cost across three standard CL benchmarks using Llama-2-7B-chat (average cost across different task orders, FTBA-MBAOI: FTBA-MBA with orthogonal initialize $\boldsymbol{A}$).

| Method | Standard CL Benchmark | | Large Number of Tasks | | SuperNI Benchmark | |
|--------|------|------|------|------|------|------|
| | Peak GPU Memory | GPU Walltime | Peak GPU Memory | GPU Walltime | Peak GPU Memory | GPU Walltime |
| O-LoRA | 35.71GB | 01:21:49 | 37.55GB | 02:47:06 | 38.06GB | 02:48:34 |
| InfLoRA | 45.21GB | 01:47:48 | 57.66GB | 04:57:22 | 58.99GB | 05:01:01 |
| FTBA-MBA | 35.29GB | 00:51:43 | 35.64GB | 01:59:51 | 36.01GB | 02:02:07 |
| FTBA-MBAOI | 35.43GB | 00:51:27 | 36.23GB | 01:59:12 | 37.59GB | 02:01:35 |
| FTBA-MB | 35.17GB | 00:51:36 | 35.47GB | 01:59:26 | 35.91GB | 02:01:49 |
| SLAO (ours) | 35.24GB | 00:50:58 | 35.61GB | 01:59:00 | 35.94GB | 02:00:08 |

## D.10 COMPARISON OF TRAINING COST

We compare the training cost among several baselines in Table 14. We use a single NVIDIA A100 GPU to fine-tune Llama-2-7B-chat. It shows that our SLAO is both memory usage efficient and training efficient, since we only compute one-time QR matrix factorization at initialization, avoiding additional computational cost during training. Besides, we observe that the walltime under orthogonal initialization is often smaller than the walltime without orthogonal initialization. There is a similar conclusion in Hu et al. (2020), which proves that drawing the initial weights from the orthogonal group can speed up convergence. Therefore, SLAO provides an ideal balance between performance, memory usage, and training speed.

## D.11 COMPARISON OF ORTHOGONALITY AMONG LORA-BASED CL METHODS

- **Initialization:**
  - (a) O-LoRA: New task's LoRA $\boldsymbol{B}_i\boldsymbol{A}_i$ is randomly initialized and is not orthogonal to previous tasks' LoRAs $\{\boldsymbol{B}_1\boldsymbol{A}_1, \ldots, \boldsymbol{B}_{i-1}\boldsymbol{A}_{i-1}\}$.
  - (b) InfLoRA: Use all of new task $i$ data to compute its input matrix $\boldsymbol{H}_i$ on current full model parameters, and use all previous $i-1$ tasks' gradient spaces which denote as $\boldsymbol{M}_i$ to make new $\boldsymbol{A}_i \in \mathbb{R}^{r \times d}$ lie in $\boldsymbol{N}_i \cap \boldsymbol{M}_i^{\perp}$ where $\boldsymbol{N}_i$ is the subspace spanned by the columns of $\boldsymbol{H}_i$. $\boldsymbol{B}_i \in \mathbb{R}^{d \times r}$ is initialized as zero.
  - (c) SLAO: Extract orthogonal basis from previous fine-tuned $\boldsymbol{A}_{i-1} \in \mathbb{R}^{r \times d}$ as new task's $\boldsymbol{A}_i$, which makes $\boldsymbol{A}_i\boldsymbol{A}_i^{\top} = \boldsymbol{I}_r$. $\boldsymbol{B}_i \in \mathbb{R}^{d \times r}$ is initialized as previous fine-tuned $\boldsymbol{B}_{i-1}$.
- **Training:**
  - (a) O-LoRA: Compute orthogonal loss to make new task's $\boldsymbol{A}_i \in \mathbb{R}^{r \times d}$ orthogonal to all previous tasks' $\boldsymbol{A}$, and update $\boldsymbol{A}_i$ and $\boldsymbol{B}_i$.
  - (b) InfLoRA: Compute standard cross entropy loss and update $\boldsymbol{B}_i \in \mathbb{R}^{d \times r}$.
  - (c) SLAO: Compute standard cross entropy loss and update $\boldsymbol{B}_i \in \mathbb{R}^{d \times r}$ and $\boldsymbol{A}_i \in \mathbb{R}^{r \times d}$.
- **Post-Training:**
  - (a) O-LoRA: Store all tasks' LoRA $\{\boldsymbol{B}_1\boldsymbol{A}_1, \ldots, \boldsymbol{B}_i\boldsymbol{A}_i\}$.
  - (b) InfLoRA: Use new task $i$ data to compute new task input matrix $\boldsymbol{R}_i$ on new learned $\boldsymbol{B}_i\boldsymbol{A}_i$, then compute new gradient orthogonal bases memory $\boldsymbol{M}_i$ through DualGPM,

where $M_i$ represents the gradient space of all $i$ tasks. Then, integrate $B_i A_i$ to $W_{i-1}$ and store the updated gradient space $M_i$ of all $i$ tasks.

(c) SLAO: Merge $B_i$ to previously merged $B_{i-1}$ and keep fine-tuned $B_i$, merged $B_i$, and fine-tuned $A_i$.

Overall, InfLoRA and SLAO both focus on the orthogonality of initialization and post-training, while O-LoRA focuses on the orthogonality of the updating process during training. Moreover, for initialization, InfLoRA makes new $A_i$ lie at the intersection of input matrix and previous gradient spaces $M_i$, while SLAO extracts orthogonal basis from previous fine-tuned $A_{i-1}$ as new $A_i$; for post-training, InfLoRA computes and stores all previous tasks' orthogonal gradient spaces, while SLAO uses the asymmetry of LoRA to obtain a merged $B$.

### D.12 DESCRIPTIONS OF TASK SEQUENCE ORDERS

We report task descriptions and their metrics used for our CL experiments across Llama models in Table 15 and Table 16. And we show eight task orders in Table 17.

Table 15: Descriptions of 15 datasets in Large Number of Tasks benchmark and first 5 datasets from standard CL benchmark.

| Dataset name | Category | Task | Domain | Metric |
|---|---|---|---|---|
| 1. Yelp | CL Benchmark | Sentiment analysis | Yelp reviews | Accuracy |
| 2. Amazon | CL Benchmark | Sentiment analysis | Amazon reviews | Accuracy |
| 3. DBpedia | CL Benchmark | Topic classification | Wikipedia | Accuracy |
| 4. Yahoo | CL Benchmark | Topic classification | Yahoo Q&A | Accuracy |
| 5. AG News | CL Benchmark | Topic classification | News | Accuracy |
| 6. MNLI | GLUE | Natural language inference | Various | Accuracy |
| 7. QQP | GLUE | Paragraph detection | Quora | Accuracy |
| 8. RTE | GLUE | Natural language inference | News, Wikipedia | Accuracy |
| 9. SST-2 | GLUE | Sentiment analysis | Movie reviews | Accuracy |
| 10. WiC | SuperGLUE | Word sense disambiguation | Lexical databases | Accuracy |
| 11. CB | SuperGLUE | Natural language inference | Various | Accuracy |
| 12. COPA | SuperGLUE | Question and answering | Blogs,encyclopedia | Accuracy |
| 13. BoolQA | SuperGLUE | Boolean question and answering | Wikipedia | Accuracy |
| 14. MultiRC | SuperGLUE | Question and answering | Various | Accuracy |
| 15. IMDB | SuperGLUE | Sentiment Analysis | Movie reviews | Accuracy |

Table 16: Descriptions of 15 datasets in SuperNI benchmark.

| Dataset number | Dataset name | Task | Metric |
|---|---|---|---|
| 1. task639 | multi-woz-user-utterance-generation | dialogue generation | Rouge-L |
| 2. task1590 | diplomacy-text-generation | dialogue generation | Rouge-L |
| 3. task1729 | personachat-generate-next | dialogue generation | Rouge-L |
| 4. task181 | outcome extraction | information extraction | Rouge-L |
| 5. task748 | glucose-reverse-cause-event-detection | information extraction | Rouge-L |
| 6. task1510 | evaluation-relation-extraction | information extraction | Rouge-L |
| 7. task002 | quoref-answer-generation | question answering | Rouge-L |
| 8. task073 | commonsenseqa-answer-generation | question answering | Rouge-L |
| 9. task591 | sciq-answer-generation | question answering | Rouge-L |
| 10. task511 | reddit-tifu-long-text-summarization | summarization | Rouge-L |
| 11. task1290 | xsum-summarization | summarization | Rouge-L |
| 12. task1572 | samsum-summary | summarization | Rouge-L |
| 13. task363 | sst2-polarity-classification | sentiment analysis | Accuracy |
| 14. task875 | emotion-classification | sentiment analysis | Accuracy |
| 15. task1687 | sentiment140-classification | sentiment analysis | Accuracy |

Table 17: Eight different task orders.

| Order | Model | Task Sequence |
|---|---|---|
| 1 | Llama-2-7B-chat, Llama-2-13B-chat, Llama-3-2-3B | dbpedia→ amazon → yahoo → ag |
| 2 | Llama-2-7B-chat, Llama-2-13B-chat, Llama-3-2-3B | dbpedia→ amazon → ag→ yahoo |
| 3 | Llama-2-7B-chat, Llama-2-13B-chat, Llama-3-2-3B | yahoo → amazon → ag → dbpedia |
| 4 | Llama-2-7B-chat, Llama-2-13B-chat, Llama-3-2-3B | mnli → cb → wic → copa → qqp → boolqa → rte →imdb → yelp → amazon → sst-2 → dbpedia → ag →multirc → yahoo |
| 5 | Llama-2-7B-chat, Llama-2-13B-chat, Llama-3-2-3B | multirc → boolqa → wic → mnli → cb → copa → qqp → rte → imdb → sst-2 → dbpedia → ag → yelp → amazon →yahoo |
| 6 | Llama-2-7B-chat, Llama-2-13B-chat, Llama-3-2-3B | yelp → amazon → mnli → cb → copa → qqp → rte →imdb→ sst-2 → dbpedia → ag → yahoo → multirc →boolqa → wic |
| 1 (SuperNI) | Llama-2-7B-chat, Llama-2-13B-chat, Llama-3-2-3B | task1572 → task363 → task1290 → task181 → task002 →task1510 → task639 → task1729 → task073 → task1590 → task748 → task511 → task591 → task1687 → task875 |
| 2 (SuperNI) | Llama-2-7B-chat, Llama-2-13B-chat, Llama-3-2-3B | task748 → task073 → task1590 → task639 → task1572 → task1687 → task591 → task363→ task1510→ task1729 → task181 → task511 → task002 → task1290 → task875 |

