# OpenReview forum: "Merge before Forget: A Single LoRA Continual Learning via Continual Merging"
_ICLR.cc/2026/Conference — ICLR 2026 Poster_

### Official Review · Reviewer_YyiC · 2025-10-31

**Soundness:** 3
**Presentation:** 3
**Contribution:** 3
**Rating:** 6
**Confidence:** 4

**Summary:**

This manuscript addresses the linear growth in memory and computational cost in LoRA-based continual learning (CL), where prior methods store a separate adapter per task. The authors propose SLAO, a novel method that maintains only a single LoRA adapter, achieving constant memory complexity. SLAO frames CL as a sequential merging problem. For each new task, it uses an orthogonal basis from the prior adapter for initialization, fine-tunes, and then performs an asymmetric merge: it replaces the task-invariant A matrix and applies a time-aware weighted average to the task-specific B matrix. Experiments on several benchmarks and Llama models show SLAO outperforms other data-free, constant-memory CL baselines.

**Strengths:**

1. The core insight to treat LoRA's A and B matrices asymmetrically is clever, well-supported by empirical analysis (Fig. 1), and validated in ablations (Table 3), proving its critical role in the method's success.

2. The method is comprehensively evaluated against the relevant baselines (adapter-isolation, merging, etc.) and is clearly shown to be the superior data-free approach. The claims are further supported by thorough ablations for initialization, merging strategy.

**Weaknesses:**

1. The merge for the A matrix is a full replacement ($A_{merge}^{i} = A_{ft,i}$) by the newest task, which risks forgetting the optimal basis for older tasks. This "winner-takes-all" approach needs more discussion.

2. A significant performance gap remains between SLAO and data-rehearsal methods, especially on complex benchmarks. A brief discussion on hybridizing SLAO's constant-memory model with a constant-sized buffer would add context.

3. The theoretical analysis provides good motivation for the design, but the final algorithm isn't a strict derivation. For instance, the scaling factor $\lambda(i) = 1/\sqrt{i}$ is adopted from prior work, not derived from the paper's own analysis.

**Questions:**

Refer to Weaknesses for related questions.

---

> ### Author Response · Authors · 2025-11-22
>
> **Q1 Explain using $\boldsymbol{A} _{\text{merge}}^{i}$ to start $\boldsymbol{A} _{ft,i}$ for new task.**
>
> Thanks for highlighting this point. We agree that a simple direct replacement of LoRA could risk forgetting previously learned subspaces. However, our proposed SLAO does not simply replace the whole LoRA. For the initialization of SLAO, we analyze it from two points:
>
> - (1) Asymmetry of LoRA components. As shown in Figure 1, $A$ matrices trained on different tasks have high similarity, which means their parameter subspaces largely overlap. This indicates that $A$ captures a shared input feature subspace that changes minimally across tasks. and thus $A$ can be reused across different tasks without significant performance degradation. In contrast, $B$ matrices differ substantially across different tasks and are nearly orthogonal to each other. Thus, if we apply the same operation on $B$, it would remove task-specific directions and lead to strong performance drops. Therefore, adopting the latest $A$ does not discard an “optimal basis”, but instead preserves the stable feature extractor learned from previous tasks.
>
>
> - (2) Initialization points in continual merging. As discussed in [1], due to model starting fine-tuning points affected by continual merging, there are three different potential starting fine-tuning points for new tasks: 1. from the pre-trained model (zero initialization for LoRA), 2. from last merging point, 3. from last fine-tuning point. In our SLAO, we utilize the last fine-tuning point $\boldsymbol{A} _{ft,i}$ as the starting point, and then we perform orthogonal initialization on $\boldsymbol{A} _{ft,i}$ to extract the orthogonal basis before training the new task, which can preserve the shared subspaces. We also conducted experiments across different benchmarks and different models to compare these three initialization points applied to SLAO. As shown in Table 2, initializing from last fine-tuning point consistently outperforms the other two initialization strategies across all three benchmarks.
>
> Thus, SLAO does not perform a 'winner-takes-all' replacement. Instead, new task's $A$ is initialized from the last fine-tuning point and then is orthogonalized. This initialization already embeds the shared subspaces learned from previous tasks, ensuring that the stable feature extractor learned across tasks is preserved rather than overwritten. This empirical stability of $A$ across tasks is the reason SLAO can safely update $A$ without degrading performance or forgetting earlier tasks. We will explore the effect of starting fine-tuning points from a theoretical perspective to better understand its role in LoRA continual merging/learning.
>
> [1] How to Merge Your Multimodal Models Over Time?, CVPR 2025.
>
> **Q2 A brief discussion on hybridizing SLAO's constant-memory model with a constant-sized buffer.**
>
> Thank you for suggesting this insightful point. We agree that rehearsal-based methods often achieve stronger performance on some complex tasks due to direct access to past task data samples. However, these methods depend on storing previous task data or generating previous pseudo data, resulting in the memory growing with the number of sequential tasks, which is often infeasible in LLM scenarios. Our goal of SLAO focuses on constant memory without storing or replaying any previous data information, and a practical data-free continual learning setting.
>
> Hybridizing SLAO with a small constant-size buffer is indeed possible. The buffer for previous data can be integrated by adding a rehearsal loss term, without modifying the framework of SLAO. But it may not be our initial goal, since we aim to propose a data-free method, which is more realistic in many LLM scenarios.

---

> ### Author Response · Authors · 2025-11-22
>
> **Q3 How does the theoretical analysis connect to the final algorithm, such as the scaling factor $\lambda(i)=1/\sqrt{i}$?**
>
> Thanks for raising this question. Our theoretical analysis explains that the orthogonal condition of $A$, that $A _{i} ^{(0)} (A _{i} ^{(0)})=I _{r}$ can minimize forgetting-intransigence error in continual learning, as shown in Eq.(6). Thus, we utilize this orthogonal initialization in our proposed SLAO. Next, we analyze the orthogonality of $B$ updates to its initialization is more than that of $A$ updates to its initialization, as shown in Eq.(10) and Eq.(11), which explains why merging $B$ is better than merging $A$. Then, since linear connectivity implies that model parameters fine-tuned from the same pretrained checkpoint can be added to improve generalization, a property that extends to PEFT adapters, whose small updates likewise allow linear composition. Hence, we use task vectors to continually merge $B$ in our proposed SLAO.
>
> We agree that time-aware scaling is adopted from prior work and remains a heuristic. However, time-aware scaling factor $\lambda(i)=1/\sqrt{i}$ is motivated by empirical observations from prior work indicating that task vectors from different tasks tend to be approximately orthogonal [1,2]. In our proposed SLAO, $B$ from different tasks are almost orthogonal to each other, as shown in Figure 1 in our manuscript, indicating that $B$ task vectors are approximately orthogonal. This orthogonality property makes  $\lambda(i)=1/\sqrt{i}$  as a natural choice for the scaling factor, as it helps maintain the magnitude of parameter changes across merging steps [3].
>
>
> We acknowledge the value of a more principled derivation would strengthen the work, and we aim to explore more adaptive and learnable scaling strategies that could generalize the time-aware heuristic and offer theoretical justification in future work.
>
> [1] Editing models with task arithmetic, ICLR 2023.
>
> [2] Parameter-efficient multi-task model fusion with partial linearization, ICLR 2024.
>
> [3] Merging on the Fly Without Retraining: A Sequential Approach to Scalable Continual Model Merging, NeurIPS 2025.

---

### Official Review · Reviewer_ucvk · 2025-11-01

**Soundness:** 2
**Presentation:** 3
**Contribution:** 2
**Rating:** 4
**Confidence:** 3

**Summary:**

This paper introduces SLAO, a novel continual learning (CL) method for large language models (LLMs) that leverages Low-Rank Adaptation (LoRA) and continual merging to maintain a single shared LoRA across tasks. Unlike existing LoRA-based CL methods that retain multiple task-specific LoRAs or generate pseudo-data, SLAO sequentially merges new task updates into a unified LoRA via orthogonal initialization and time-aware scaling. The approach ensures constant memory usage, reduces task interference, and improves generalization. Theoretical analysis and extensive experiments on multiple benchmarks and model scales (e.g., Llama-2, Llama-3) demonstrate its effectiveness and efficiency.

**Strengths:**

+ Provides a formal analysis of forgetting and intransigence in the NTK regime and motivates the design via LoRA asymmetry.
+ The concept of continual merging into a single LoRA is novel and addresses key limitations of existing LoRA-based CL methods.

**Weaknesses:**

+ The method is designed for llm but evaluated on models that are not state-of-the-art, as well as tasks that are easy for current llms. Models such as qwen2.5/3 series and tasks such as aime, livecodebench or at the same difficulty level are needed. At lease, the reviewer think the tasks should be more diverse.
+ Could the orthogonal initialization strategy be combined with other PEFT methods?

**Questions:**

See the weakness

---

> ### Author Response · Authors · 2025-11-22
>
> **Q1 How does proposed method perform using state-of-the-art LLMs, like qwen2.5/3, and complex tasks?**
>
> Thanks for suggesting evaluating our proposed method SLAO, using state-of-the-art LLMs and more challenging tasks. To provide a more comprehensive validation, we conducted additional experiments using Qwen2.5-3B and Qwen2.5-7B models on MATH, LiveCodeBench (code generation lite), and SuperNI benchmarks.
>
> (1) MATH $\leftrightarrow$ LiveCodeBench: two-task continual learning
>
> For MATH and LiveCodeBench, we created two task orders:
> - Order 1: MATH (geometry/counting and probability)$\rightarrow$LiveCodeBench
> - Order 2: LiveCodeBench$\rightarrow$MATH (geometry/counting and probability).
>
> Since LiveCodeBench (code generation lite) only has public testing data, we follow standard practice and split it into 80\% testing data as our training data, and 20\% testing data as our testing data. We use the geometry and counting-and-probability datasets in MATH since their dataset sizes are comparable to the size of LiveCodeBench. We compared SLAO with the other two LoRA-based continual learning methods.
>
> Experiments with Qwen2.5-7B on two task orders across MATH (geometry) and LiveCodeBench evaluated via metric pass@1:
>
> | Method  | Order 1   | Order 2   | Avg  |
> |:--------|-----:|------:|-----:|
> | O-LoRA  | 26.11|25.80|25.96|
> | InfLoRA | 26.72|26.27|26.50|
> | SLAO    | 26.87|26.89|26.88|
>
> Experiments with Qwen2.5-3B on two task orders, where  Order 1 is MATH (counting and probability)$\rightarrow$LiveCodeBench, and Order 2 is LiveCodeBench$\rightarrow$MATH (counting and probability), evaluated via metric pass@1:
>
> | Method  | Order 1   | Order 2   | Avg  |
> |:--------|-----:|-----:|-----:|
> | O-LoRA  | 21.22|11.15|16.19|
> | InfLoRA | 23.08|18.77|20.93|
> | SLAO    | 25.38|27.09|26.24|
>
> For two orders of MATH and LiveCodeBench using Qwen2.5-3B and Qwen2.5-7B, SLAO performs consistently better than other two methods. We will explore larger-scale and more diverse math–code task sequences in future work.
>
> (2) SuperNI benchmark: a long sequence of tasks continual learning
>
> Experiments with Qwen2.5-3B on two task orders of SuperNI benchmark via metrics in the manuscript:
>
> | Method  | O1 | O2 | Avg |
> |---------|-------:|-------:|--------:|
> | O-LoRA  | 31.1  | 29.8  | 30.5   |
> | InfLoRA | 35.6  | 25.6  | 30.6   |
> | SLAO    | **37.8** | **32.4** | **35.1** |
>
>
> Experiments with Qwen2.5-7B on two task orders of SuperNI benchmark via metrics in the manuscript:
>
> | Method  | O1 | O2 | Avg |
> |---------|-------:|-------:|--------:|
> | O-LoRA  |34.3  | 32.6  | 33.4   |
> | InfLoRA | **43.5** | 31.0  | 37.3   |
> | SLAO    | 41.0 | **35.5** | **38.3** |
>
>
>  When evaluating SuperNI benchmarks, SLAO is distinctively better than baselines and achieves the best average performance on both Qwen2.5-3B and Qwen2.5-7B models, demonstrating its robustness on model sizes. We will add experimental results on Qwen models in the further revised version.
>
>
> **Q2 Could the orthogonal initialization strategy be combined with other PEFT methods?**
>
> Thanks for raising this insightful question. Our orthogonal initialization strategy is derived from the intrinsic structural asymmetry of LoRA components in forgetting-intransigence error analysis. We find that making $A _{i} ^{(0)}(A _{i} ^{(0)}) ^{\top} = I _r$ for $i-$th task can minimize the forgetting-intransigence error in Eq.(6).
>
> Other PEFT methods, such as prompt tuning and prefix tuning, do not provide a pair of learnable low-rank matrices. Their parameter spaces are not structured in a way like LoRA. We believe our orthogonal initialization cannot be directly extended to other PEFT methods in a meaningful way. That said, extending our orthogonalization to other PEFT methods would require redefining their parameterization to support orthogonal subspaces like LoRA. This would be a challenging design and is an interesting direction for future work, but it would require new formulations rather than a direct integration of SLAO’s initialization.

---

### Official Review · Reviewer_TKgi · 2025-11-01

**Soundness:** 3
**Presentation:** 2
**Contribution:** 3
**Rating:** 6
**Confidence:** 3

**Summary:**

This paper proposes SLAO (Single LoRA Continual Learning with Orthogonal Initialization via Continual Merging), a framework that maintains a single shared LoRA across all tasks by sequentially merging new task updates, eliminating the need for task-specific LoRA storage.  Experiments on Llama-2-7B-chat, Llama-2-13B-chat, and Llama-3-3B demonstrate SLAO’s effectiveness.

**Strengths:**

1. SLAO is the first to enable CL with a single shared LoRA via sequential merging.

2. SLAO is robust to hyperparameters (e.g., LoRA rank, learning rate) and model scales. In particular, performance improves with larger models.

**Weaknesses:**

1. While SLAO’s training overhead is low, QR decomposition for orthogonal basis extraction adds a one-time cost per task. The paper does not quantify this cost for long sequences (e.g., 50+ tasks) or analyze whether approximate orthogonal methods (e.g., randomized SVD) could reduce the cost without performance loss.

**Questions:**

1. How does SLAO perform when tasks have extreme similarity (e.g., multiple sentiment analysis tasks) or dissimilarity (e.g., sentiment vs. QA)? Does orthogonal initialization overly restrict knowledge transfer for similar tasks, and can the orthogonality constraint be relaxed dynamically?

---

> ### Author Response · Authors · 2025-11-22
>
> **Q1 QR cost quantification in SLAO and compare orthogonal initialization in SLAO with approximate orthogonal methods (e.g., randomized SVD)**
>
> Thanks for suggesting quantifying the cost caused by QR decomposition in SLAO. To address this, we compared the training cost when using different orthogonal methods applied to SLAO when using Llama-2-7B-chat on three continual learning widely used benchmarks. Note that (1)  SLAO (Randomized SVD) performs Randomized SVD on $A$ and utilizes $QU$ as new orthogonal $A$, (2) SLAO (SVD) performs rank-$k$ SVD on $A$, and utilizes $UV ^{\top}$ as new orthogonal $A$. All experiments follow the same setup as in the original manuscript.
>
> (1) Comparison of training cost:
>
> | Method       |      Standard Benchmark     |                   |     Long Sequence         |                   |       SuperNI               |                   |
> |:------------:|:---------------------------:|:-----------------:|:--------------------------:|:-----------------:|:----------------------------:|:-----------------:|
> |          |     Peak GPU Mem.         |  GPU Walltime     |     Peak GPU Mem.        |  GPU Walltime     |     Peak GPU Mem.          |  GPU Walltime     |
> | O-LoRA   |          35.71 GB           |     01:21:49      |          37.55 GB          |     02:47:06      |          38.06 GB            |     02:48:34      |
> | SLAO (Randomized SVD)  |35.74 GB | 01:02:33|37.36 GB|02:10:59 |37.66 GB |02:03:23 |
> | SLAO (SVD)     |    35.72 GB     |  01:02:16     |   36.10 GB     |   02:01:16        |   36.37 GB      |   02:03:08       |
> | SLAO (QR)     |          35.24 GB           |     00:50:58      |          35.61 GB          |     01:59:00      |          35.94 GB            |     02:00:08      |
>
>
> From these results, randomized SVD costs slightly more memory compared to rank-$k$ SVD and QR, since randomized SVD has both a QR factorization and an SVD operation. Rank-$k$ SVD and QR have similar memory footprints, but QR is consistently faster.
>
> (2) Comparison of performance:
>
> | Method   |      Standard Benchmark     |       |        |        |     Long Sequence   |       |        |         |               SuperNI               |       |        |
> |:--------:|:--------:|:--------:|:--------:|:--------:|:--------:|:--------:|:--------:|:--------:|:--------:|:--------:|:--------:|
> |          |     O1         |  O2    |     O3        |  Avg    |     O1          |  O2    | O3 | Avg | O1 |O2| Avg|
> | O-LoRA   |76.1|76.3|79.2|77.2|74.0|72.0|74.6|73.5|23.3|28.4|25.9|
> | SLAO (Randomized SVD) |72.6|69.1|73.3|71.7|52.3|62.5|57.8|57.5|11.8|22.8|17.3|
> | SLAO (SVD)    |79.9|80.8|80.2|80.3|75.3|74.4|75.1|74.9|36.9|33.7|35.3|
> | SLAO (QR)     |80.1|80.8|80.4|80.4|75.0|74.4|75.1|74.8|38.7|35.7|37.2|
>
>
> The performance of Randomized SVD is not better than rank-$k$ SVD and QR. Rank-$k$ SVD and QR have similar results on standard CL benchmark and long sequence benchmark, but the performance of QR on SuperNI benchmark is better than that of rank-$k$ SVD. Thus, we choose QR as our orthogonal initialization method for SLAO, providing a good balance between computational cost and performance. We will include the results of randomized SVD in the further revised version.

---

> ### Author Response · Authors · 2025-11-22
>
> **Q2 How does SLAO perform when tasks have extreme similarity and dissimilarity? Can orthogonal constraint be relaxed dynamically?**
>
> Thanks for raising this insightful question.
>
> (1) To evaluate SLAO performance on tasks having extreme similarity, we choose four sentiment analysis tasks, including imdb, yelp, amazon, and sst-2. We create three different orders across these four tasks and use Llama-2-7B-chat to compare our SLAO with other continual learning methods.
>
>
> | Method  | O1   | O2   | O3   | Avg  |
> |:--------|-----:|-----:|-----:|-----:|
> | O-LoRA  | 65.9|68.6|68.4|67.6|
> | InfLoRA | 71.1|70.0|69.3|70.1|
> | SLAO    | 69.4|71.8|69.8|70.3|
>
> The results show that InfLoRA and SLAO has similar performance when tasks have extreme similarity, evidently better than O-LoRA. It indicates orthogonal initialization does not overly restrict knowledge transfer for similar tasks.
>
> (2) To evaluate SLAO performance on tasks having dissimilarity, we choose two sentiment analysis tasks (amazon and sst-2) and two QA tasks (COPA and MultiRC). We create three different orders across these four tasks and use Llama-2-7B-chat to compare our SLAO with other continual learning methods.
>
> | Method  | O1   | O2   | O3   | Avg  |
> |:--------|-----:|-----:|-----:|-----:|
> | O-LoRA  |62.4|60.8|64.4|62.5|
> | InfLoRA |62.6|62.2|61.0|61.9|
> | SLAO    |64.1|62.1|63.7|63.3|
>
> For tasks which have extreme dissimilarity, SLAO has better average results and smaller variance across three orders than other two baselines. It shows that SLAO is more robust for conflicting task LoRA gradients and has a strong ability to reduce interference.
>
> Based on these experimental results, when tasks have extreme similarity, SLAO and InfLoRA have similar performance, and when tasks have extreme dissimilarity, SLAO has better performance than other two baselines. For SLAO itself, it has stronger performance when tasks have extreme dissimilarity than when tasks have extreme similarity.
>
> For dynamic relation of orthogonality, while SLAO does not explicitly relax orthogonality during training, the optimization process implicitly adapts task-specific low-rank subspaces, and the post-training merging step improves LoRA generalization since it merges LoRA weights of each sequential task. It's a promising future direction to extend SLAO with dynamic and explicitly soft orthogonal constraints, and we appreciate the reviewer's suggestions and would like to explore it in the revised version.

---

### Official Review · Reviewer_tRRS · 2025-11-01

**Soundness:** 2
**Presentation:** 3
**Contribution:** 2
**Rating:** 4
**Confidence:** 4

**Summary:**

This paper proposes SLAO, an algorithm that can avoid forgetting while maintaining a single shared LoRA across all tasks, avoiding the linear memory growth of storing multiple LoRAs. The authors provide NTK-based theoretical analysis to motivate orthogonality and asymmetry handling, and show consistent improvements over LoRA-based CL methods.

**Strengths:**

+ The motivation of the intrinsic asymmetry property of LoRA is clearly presented in Figure 1.
+ The paper provides theoretical analyses grounded in NTK theory.
+ The writing is generally clear and well-organized.

**Weaknesses:**

+ Unclear mechanism for avoiding forgetting. 1)While I can understand how InfLoRA prevents forgetting by projecting updates into the null space of old task features or directly using old task samples, I find it difficult to see how the proposed orthogonal initialization in this paper achieves the same effect. For the merging step, many prior works in multi-task learning compute task vectors as ΔW = B·A. In your formulation, however, A is replaced, and only B is merged. It remains unclear why merging B alone can effectively prevent forgetting. 2) Another concern is, the definition of task vector here is different from that in MTL, where the fine-tuned parameters subtract the same original weights instead of the merged ones. 3) It would be helpful to explore the effect of the scaling factor $\lambda(i)$.
+ The paper claims that the proposed method avoids linear memory growth by maintaining only the merged LoRA and the current task’s LoRA, without storing all task-specific adapters. However, in methods such as O-LoRA and InfLoRA, it is also possible to directly merge different task-specific LoRAs while achieving comparable performance. I would appreciate clarification on how Figure 2 was computed in this regard, and a more detailed explanation of how the proposed approach fundamentally differs from these existing merging strategies in terms of scalability.
+ While analytically clean, this setting does not fully capture nonlinear behavior in LLM fine-tuning. The link between the NTK bounds and empirical performance could be better validated with empirical NTK measurements.

**Questions:**

Please refer to the weaknesses.

---

> ### Author Response · Authors · 2025-11-22
>
> **Q1.1 How does proposed orthogonal initialization work? Why can merging B alone effectively prevent forgetting?**
>
> Thanks for raising this important question. Our proposed asymmetric orthogonal initialization and time-aware scaling merging strategies stem from the way the asymmetric structure of the LoRA components contributes to forgetting–intransigence error in continual learning, as derived in our analysis of Eq.(6) on Page 4, and how $A$ and $B$ themselves exhibit different orthogonality properties, as illustrated in Figure 1.
>
> (1) For why orthogonal initialization of $A$ can reduce forgetting-intransigence error in CL, we derived results from the decomposition of forgetting error and intransigence error, as shown in Eq.(6) on Page 4. Specifically, to minimize the errors in Eq.(6), we can minimize both terms $\\|A _{t}-A _{i}\\| _{F}$ and $\\|A _{i}-A _{i} ^{*}\\| _{F}$ in error decomposition if we make $A _{i} ^{(0)}(A _{i} ^{(0)}) ^{\top} = I _r$ where $i\in[1,\dots,t]$, which makes a balance between these two error terms. This orthogonality not only keeps geometric consistency across tasks but also allows $A _{j} (t\geq j>i)$ to remain well-aligned with previous $A _{i}$, i.e. $\mathbb{E}[A _{j}A _{i} ^{\top}]\approx I _{r}$, thereby minimizing the errors.
>
>
> (2) The effect of merging $B$ is based on the orthogonality of $B$ in LoRA, as $B$ trained on different tasks tends to be more orthogonal than $A$, as shown in Figure 1. We also analyzed the orthogonality of $A$ and $B$ updates to their initializations analytically, as shown in Eq.(10) and Eq.(11), respectively. We find that $\\|\Delta B _{i} ^{\top} B _{i-1}\\| _{F}<\\|A _{i-1}\Delta A _{i} ^{\top}\\| _{F}$, which means the update of $B$ to its initialization tends to be more orthogonal than the update of $A$ to its initialization. Then, following the findings in [1], task vectors used in model merging are inherently nearly orthogonal in order to minimize interference, which indicates that in our case, merging $B$ rather than merging $A$ provides better task isolation and reduced interference, thus merging $B$ can reduce forgetting.
>
> [1] Modeling multi-task model merging as adaptive projective gradient descent, ICML 2025.
>
> **Q1.2 Definition of task vector is different from that in MTL.**
>
>
> Thanks for raising this question. The key difference between our CL setting and the MTL setting lies in the starting point for training each task. As discussed in [1], traditional model merging in MTL setting assumes that all task models are fine-tuned from the same pre-trained model, but in CL setting, the number of potential starting points for fine-tuning a new task increases as tasks arrive. This means there are several possible choices for where to begin training a new task: (1) from the base weights (as in traditional merging), (2) from a merged combination of previous tasks, or (3) from last fine-tuning point. Thus, $t-$th continual task vector can be defined as $\tau _{t} =\theta _{ft} ^{t} - \theta _{t-1}$ [1], which means the starting training point $\theta _{t-1}$ can be the last fine-tuning/merging point. We also discussed the performance of different starting/initialization points for new tasks in our proposed method, as shown in Table 2 on page 8. It shows that initializing from last fine-tuning point consistently outperforms other two strategies across all three benchmarks, and using last merging point performs slightly worse, while random (zero) initialization performs the worst. Since we merge $B$ in SLAO and $B$ is initialized as zero for the first sequential task, the difference between starting from the last fine-tuning point and the last merging point is that last merging point fixes time coefficients after merging back to a single LoRA, limiting its flexibility, while initialization from last fine-tuning point allows the merged LoRA to implicitly re-weight previous tasks’ updates when merging.
>
>
> [1] How to Merge Your Multimodal Models Over Time?, CVPR 2025.
>
> **Q1.3 The effect of the scaling factor $\lambda(i)$.**
>
> Thanks for asking this question. We previously showed the effect of scaling factor  $\lambda(i)$ in Figure 3 on Page 9. We compared it against fixed factors $\{0.1, 0.5, 0.9\}$ on standard CL benchmark, large number of tasks, and SuperNI benchmarks, via Llama-2-7B-chat and Llama-3-2-3B. As shown in Figure 3, adaptive strategy consistently achieves highest average accuracy with lower variance across task orders and models. For simpler standard CL benchmark, larger fixed value 0.9 outperforms a smaller one 0.1, while for more complex or long benchmarks, smaller values perform better; 0.5 is relatively stable but consistently suboptimal.

---

> ### Author Response · Authors · 2025-11-22
>
> **Q2.1 How is Figure 2 computed?**
>
> We appreciate this question. Figure 2 compares the memory usage of SLAO and O-LoRA as the number of tasks increases. In O-LoRA algorithm, it explicitly defines the orthogonal loss term as $\sum _{i=1} ^{t-1}\mathcal{L} _{orth}(A _i, A _t)$, where $\mathcal{L} _{orth}(A _i, A _t)=\sum _{j,k}\\| (A _{i} ^{\top} A _{t}) _{j,k}\\| ^{2}$. This orthogonal loss term requires a new task $t$ LoRA to compute the orthogonality between each previous task's LoRA, thus during new task training, previous LoRA parameters $\\{A _{i}, B _{i}|i<t\\}$ are frozen and are used to participate in training loss computation rather than merged. Therefore, **in Figure 2**:
>
> - when training the second task, SLAO has $\\{\text{LoRA} _{\text{merged}}, \text{LoRA} _{2}\\}$, which are one previously merged LoRA and one new task LoRA, and O-LoRA has $\\{\text{LoRA} _{1}, \text{LoRA} _{2}\\}$, which are one previously fine-tuned LoRA and one new LoRA;
> - when training the third task, SLAO has  $\\{\text{LoRA} _{\text{merged}}, \text{LoRA} _{3}\\}$, which are still one previously merged LoRA and one new task LoRA, while O-LoRA has $\\{\text{LoRA} _{1}, \text{LoRA} _{2}, \text{LoRA} _{3}\\}$, which are two fine-tuned LoRA and one new task LoRA;
> - when training $t-$th task, SLAO has $\\{\text{LoRA} _{\text{merged}}, \text{LoRA} _{t}\\}$, while O-LoRA has $\\{\text{LoRA} _{1},\dots, \text{LoRA} _{t}\\}$
>
> In summary, the memory usage of O-LoRA linearly grows with the number of tasks, while SLAO keeps constant memory usage regardless of the number of tasks, as shown in Figure 2.
>
> Regarding "possible to directly merge different task-specific LoRAs while achieving comparable performance", different merging strategies affect the performance of the merged model. There are two main classes of LoRA merging [1]: Linear merging of LoRA ($\Delta W = (\alpha _1 B _1 + \alpha _2 B _2$)($\alpha _1 A _1 + \alpha _2 A _2$)) and Concatenation of LoRA ($\Delta W = \alpha _1 B _1 A _1 + \alpha _2 B _2 A _1$). Concatenation of LoRA adds a step that trains the merging coefficients on a small mixture of datasets, when different tasks' LoRA are trained [1]. For linear merging of LoRA, LoRA Hub [2], TIES [3], DARE [4] learns static $\alpha$ values for every layer to reduce different LoRA interference. However, InfLoRA only mentioned that after training $t$-th task, their LoRA branch can be integrated into the pre-trained model by simply $W _t = W _{t-1} + B _{t}A _{t}$ without considering how the merged task-specific LoRA affects catastrophic forgetting of pre-trained model knowledge if directly merging back to the pre-trained model, as discussed in [5]. As for O-LoRA, their merging operation happens after training all existing downstream tasks since their algorithm requires the new task to see all previous tasks LoRA, as discussed before.
>
> [1] LoRA Soups: Merging LoRAs for Practical Skill Composition Tasks, COLING 2025.
>
> [2] LoraHub: Efficient Cross-Task Generalization via Dynamic LoRA Composition, COLM 2024.
>
> [3] Ties-merging: Resolving interference when merging models, NeurIPS 2024.
>
> [4] Language Models are Super Mario: Absorbing Abilities from Homologous Models as a Free Lunch, ICML 2024.
>
> [5] CorDA: Context-Oriented Decomposition Adaptation of Large Language Models for Task-Aware Parameter-Efficient Fine-tuning, NeurIPS 2024.

---

> ### Author Response · Authors · 2025-11-22
>
> **Q2.2 How does proposed approach fundamentally differ from these existing strategies in terms of scalability?**
>
> Thank you for this question. SLAO differs from existing merging strategies in two fundamental aspects:
>
> (1) Continual merging vs. Traditional (one-shot) merging.
>
> Existing LoRA merging methods, including KnOTS [1] and LoRA-LEGO [2], assume a single merging point where all task-specific LoRAs are concurrently accessed and are all fine-tuned from the same pre-trained model. For these methods:
> - memory usage grows linearly with the number of tasks,
> - concurrent access to all task models limits their applicability to the continual merging scenarios where past LoRAs cannot be stored or revisited.
>
> However, in SLAO, merging points are continually changed when new tasks arrive since we merge each new LoRA immediately into the previously merged LoRA to obtain a new merged LoRA. This means SLAO only needs to maintain (1) one merged LoRA and (2) one fine-tuned LoRA, instead of storing or revisiting all previously learned LoRAs. As a result, SLAO has a constant memory usage regardless of the number of tasks, enabling scalability to long task sequences in a memory-efficient way.
>
> (2) Leveraging the Asymmetry of LoRA components.
>
> Prior merging approaches do not distinguish between the roles of LoRA components $A$ and $B$, and typically either (1) merge the full task vector $BA$ or (2) merge $B$ and $A$ in the same way. However, prior works about LoRA property, such as [3], have shown that $A$ extracts features from the input, whereas $B$ uses these features to create the desired output. Thus, we further investigated this asymmetry, and as shown in Figure 1, $A$ exhibits significantly higher similarity across tasks compared to $B$, suggesting that LoRA components follow inherently different learning dynamics. This asymmetry further affects the different orthogonalities of $B$/$A$ update to its initialization, as shown in Eq.(10) and Eq.(11). Thus, given that parameter-efficient module linear arithmetic composition improves generalization, we use task vectors to continually merge $B$ in a time-aware scaling operation, instead of merging both LoRA components.
>
>
> [1] Model merging with SVD to tie the knots, ICLR 2025.
>
> [2] Merging loRAs like playing LEGO: Pushing the modularity of loRA to extremes through rank-wise clustering, ICLR 2025.
>
> [3] Asymmetry in low-rank adapters of foundation models, ICML 2024.
>
>
> **Q3 NTK bounds in LLM fine-tuning**
>
> Thank you for raising this insightful comment. We agree that NTK-based analysis is an idealized view and does not fully capture all nonlinear dynamics of LLM fine-tuning. Our goal is not to claim that SLAO operates strictly within the NTK regime, but rather to use the NTK perspective as a theoretical lens to explain why orthogonal initialization can minimize forgetting-intransigence error in continual learning.
>
> Our analysis of fine-tuning LLM via LoRA follows recent theoretical analysis in [1,2], which establish that
>
> - [1] presents theoretical guarantees on the trainability and generalization capabilities of LoRA fine-tuning of pre-trained models.
>
> - [2] provides empirical and theoretical evidence that prompt-based and low-rank adaptation stays within the NTK regime during fine-tuning.
>
> Measuring the exact empirical NTK for billion-level LLMs is computationally infeasible, as it requires computing and storing Jacobians with respect to model parameters. However, we provide indirect empirical evidence consistent with NTK predictions, where orthogonal initialization in SLAO improves parameter updates and minimizes continual learning errors. This aligns with the qualitative predictions of the NTK perspective. The NTK analysis works as a motivation that explains the design of SLAO, while full empirical NTK reconstruction for LLMs is not tractable. Our empirical results support the qualitative predictions of the theory.
>
>
> [1] LoRA Training in the NTK Regime has No Spurious Local Minima, ICML 2024.
>
> [2] A kernel-based view of language model fine-tuning, ICML 2023.

---

### Meta-Review · Area_Chair_q62D · 2026-01-04

**Summary:**

The paper introduces a new continual learning method that addresses the challenge of linear memory growth in LoRA-based merging methods. Reviewers agree the proposed constant-memory CL method with a single LoRA is novel. The asymmetric treatment of the LoRA A/B matrices is well motivated and supported by empirical results.
The method is technically sound and the experiments consistently outperform data-free baselines while maintaining constant memory.

In the rebuttal, the authors addressed the core concern of evaluation depth by adding experiments on SOTA models and complex tasks. While there remain concerns that NTK analysis is idealized and the scaling factor is heuristic, the work is solid and makes meaningful contributions. Therefore, the decision is to accept.

**Reviewer Concerns:**

The rebuttal addresses tRRS's concerns on forgetting mechanism, task vector definition, scaling factor effect, Figure 2 computation, scalability differences, and NTK limitations via detailed explanations and references. The remaining concern is that the NTK analysis remains an idealized view and may not adequately capture the non-linear LLM dynamics.

It addresses TKgi's QR cost concern with new experiments comparing costs and performance of orthogonal methods including QR, SVD, randomized SVD, and extreme task similarity/dissimilarity concern via additional benchmarks showing robustness.

For ucvk, it addresses model and task limitations with new results on Qwen2.5 models and harder tasks, including MATH, LiveCodeBench and SuperNI benchmark, demonstrating consistent gains. Integration with other PEFT methods is explained as not applicable without significant redesign.

It addresses YyiC’s concerns about the replacement of $A$ through asymmetry analysis and ablations on initialization points. Outstanding concerns include that the derivation of the scaling factor is still heuristic and the gap to rehearsal-based methods.

**Reviewer Scores:**

Based on the analysis in Reviewer Concerns, Reviewer tRRS and ucvk would likely increase from 4 to 6, resulting in all positive scores. Reviewer TKgi would likely stay at 6 or might increase to 8. Reviewer YyiC is likely to remain unchanged at 6.

---

### Decision · Program_Chairs · 2026-01-26

Accept (Poster)